# Structure and interactions of the endogenous human Commander complex

Saara Laulumaa[1,2], Esa-Pekka Kumpula [1,2], Juha T. Huiskonen [1]✉ & Markku Varjosalo [1]✉

The Commander complex, a 16-protein assembly, plays multiple roles in cell homeostasis, cell cycle and immune response. It consists of copper-metabolism Murr1 domain proteins (COMMD1–10), coiled-coil domain-containing proteins (CCDC22 and CCDC93), DENND10 and the Retriever subcomplex (VPS26C, VPS29 and VPS35L), all expressed ubiquitously in the body and linked to various diseases. Here, we report the structure and key interactions of the endogenous human Commander complex by cryogenic-electron microscopy and mass spectrometry-based proteomics. The complex consists of a stable core of COMMD1–10 and an effector containing DENND10 and Retriever, scaffolded together by CCDC22 and CCDC93. We establish the composition of Commander and reveal major interaction interfaces. These findings clarify its roles in intracellular transport, and uncover a strong association with cilium assembly, and centrosome and centriole functions.

The recently discovered multiprotein Commander complex[1–5] is a master regulator of cellular homeostasis within endosomal transport and the trans-Golgi network[6,7]. The Commander complex has been linked to the regulation of several cellular functions, including ion and lipid homeostasis[8–10], embryogenesis[6], immune response[5,11–15], cell growth[16] and cell cycle[15,17–19]. It consists of ten homologous copper-metabolism Murr1 domain proteins (COMMD1–10), coiled-coil domain-containing proteins CCDC22 and CCDC93, vacuolar protein sorting-associated proteins VPS26C, VPS35L and VPS29 (refs. 7,20), and a DENN (differentially expressed in normal and neoplastic cells) domain-containing protein 10 (DENND10)[4,21]. The complex regulates cell signaling by modulating trafficking of transmembrane channel proteins and receptors to the cell surface and diverting them from lysosomal degradation.

COMMD proteins 1–10 are highly conserved sequence homologs of COMMD1 found in mammals and some in lower metazoans. COMMD1 was named after its initially discovered function in copper homeostasis regulation[22]. Knockouts of individual COMMD proteins cause severe reduction in protein levels of all COMMD proteins, suggesting that they assemble into a larger complex[23]. Knockdowns of COMMD proteins have been associated with developmental disorders in lower vertebrates[6], while in humans, missense mutations in the

COMMD genes have been linked to various human diseases such as Wilson's disease[24], Parkinson's disease[25], atherosclerosis[23], as well as viral protein recycling[20,26] and cancer[27–29].

The CCDCs belong to a large family of CCDC proteins that have wide-ranging roles in the cell including protein trafficking, ciliary motility and cell division[20,30]. Both CCDC22 and CCDC93 contain an N-terminal microtubule binding NDC80 and NUF2–calponin-homology (NN–CH) domain and a C-terminal coiled-coil region. In particular, mutations in CCDC22 have been associated with the Ritscher–Schinzel syndrome (RSS)[13,31]. DENND10 is a member of the DENND protein family of guanine nucleotide exchange factors targeting Rabs[32]. DENND10 is involved in late endosome homeostasis and exosome biogenesis and has been proposed to interact with Rab27 (ref. 33).

The Commander complex is linked to two other multiprotein complexes: the Retriever and the Retromer. The Retriever is a recently discovered heterotrimeric complex composed of VPS29, VPS35L and VPS26C (ref. 20). It is thought to play a role similar to, but separate from, the homologous Retromer, which acts as a master controller of cargo sorting in eukaryotes. Both Retromer and Retriever interact with the heteropentameric Wiskott–Aldrich syndrome proteins and SCAR homolog (WASH) complex, an endosome-specific Arp2/3 complex

[1]Institute of Biotechnology, Helsinki Institute of Life Science HiLIFE, University of Helsinki, Helsinki, Finland. [2]These authors contributed equally: Saara Laulumaa, Esa-Pekka Kumpula. ✉e-mail: juha.huiskonen@helsinki.fi; markku.varjosalo@helsinki.fi

activator that induces actin patch formation on endosomes[20,34,35]. Despite several studies focusing on individual Commander subunits, further systematic studies that assess the complete complex are necessary to understand how the multitude of its different functions are regulated. Similarly, the structure of the Commander complex, essential for understanding its molecular function, has remained elusive.

In this study, we address these critical gaps in our knowledge of the Commander complex. We report the structure of the endogenous human Commander complex determined by cryogenic-electron microscopy (cryo-EM) and comprehensively map the molecular context and interactions of the Commander and individual complex interactions in human cells using affinity purification with mass spectrometry (AP–MS) and proximity labeling biotin identification (BioID). These findings provide a structure-based blueprint for understanding the function of the Commander complex in endosomal sorting and other essential cellular processes.

## Results

### Commander assembles into a hetero-hexadecameric complex

To investigate the structure and molecular context of the endogenous human Commander complex, we used a systematic AP–MS and BioID–MS workflow[36] to identify stable and transient protein–protein interactions (PPIs) and combined it with cryo-EM structure determination (Fig. 1a). We analyzed 14 individual Commander complex components tagged with MAC-tags[36] at either the N- or C-terminus, totaling 19 bait proteins (Fig. 1b).

Using AP–MS, we identified 69 stably interacting proteins with 390 high-confidence interactions (HCIs) for the 19 bait proteins (Fig. 1c and Supplementary Data 1) and captured 93% of all the possible binary interactions within the Commander complex (Fig. 1c and Supplementary Data 1). A comparison of the detected interactions with six interaction databases (BioGRID, IntAct, PINA2, String, bioplex and human cell map) revealed that approximately 25% (115 interactions) were previously unreported (Extended Data Fig. 1 and Supplementary Data 1).

Many of the interactions were detected with C-terminally tagged CCDC22 (Extended Data Fig. 1b). A Reactome pathway analysis of these proteins showed a more than 20-fold enrichment of the R-HSA-5617833-Cilium Assembly pathway from OFD1, PCM1, NDE1, HAUS4, EXOC4, EXOC6, EXOC5, HAUS5, HAUS1 and DZIP1 (Supplementary Data 2). A role in cilia modulation is supported by COMMD3, -9 and -10 interactions with gelsolin (GSN), an actin regulator involved in ciliogenesis[37]. Moreover, Guanine nucleotide-binding protein G(s) subunit α (GNAS), which interacts with COMMD3–5, -9 and -10, has been recently implicated in ciliogenesis in renal cells[38].

Of the cilial proteins, PCM1 (Pericentriolar Material 1)[38], MTMR2 (Myotubularin-related protein), TRIM27 (ref. 39) and Exocyst complex proteins EXOC4–6 are WASH complex recruiting proteins (Supplementary Data 1). The PCM1 anchors the WASH complex at the centrosome[40], while MTMR2 modulates the levels of phosphatidylinositol 3-phosphate (PI(3)P) on endosomes, recruiting the WASH complex[21,41]. The Exocyst complex and DZIP1 recruit the WASH complex via Rab8 and Rab11 (refs. 42–44). CCDC22 directly interacts with WASHC2A, while WASHC1 homolog WASH6P is a HCI for CCDC22, CCDC93 and DENND10. Additionally, CCDC93 participates in vesicular transport via MICAL3 (ref. 45), while CCDC22 is involved in this process via VPS33A, VPS33B, VIPAS39 (refs. 46,47) and NDR1 (ref. 48). Finally, CCDC22 interacts with microtubule organizing CAMSAP2 (Calmodulin-regulated spectrin-associated protein 2), which functions in dendrite development[49].

From the proteins interacting with COMMD1–10 (Supplementary Data 2), a Reactome pathway analysis extracts a six-protein cluster of R-HSA-6798695-Neutrophil degranulation (s100a7, s100a8, DSC1, DSG1, DSP, GSN) in line with a recently found link to platelet granulation[50]. COMMD1, -2 and -6 interact with Desmosomal proteins DSC1, DSG1 and DSP that constitute the major adhesive components of desmosomes linked to the cytoskeleton and vesicular traffic via the

Exocyst complex[51,52]. COMMD4, VPS26C and VPS35L interact with SPECC1L, an adherens junction protein that regulates cell motility and adhesion[53]. S100a7 and s100a8 are inflammation regulating proteins that also regulate NF-kB signaling[54]. Inflammation is also regulated by FKBP5 (interactor of CCDC93 and DENND10) that activates Akt and NF-kB signaling pathways[55,56]. We also detected previously reported interactions from COMMD1 to cell cycle regulating transcription factors E2F6 and TFDP1. Finally, we analyzed hierarchical clustering of the normalized relative copy amounts of Commander components based on their similarity in interaction abundances, as shown in Fig. 1d.

### CCDC93 and CCDC22 interweave within the COMMD-ring

We determined cryo-EM density maps of native as well as crosslinked Commander complex at nominal resolutions of 3.3 and 2.9 Å, respectively (Table 1, Fig. 2, Extended Data Figs. 2d,e and 5c and Supplementary Information). The crosslinked density showed improved features compared to the native dataset and was used for model building.

The COMMD-ring forms the highly interconnected core of the complex (Fig. 3, Extended Data Fig. 3a–d and Supplementary Information). It can further be divided into two halves based on the locations of the CCDCs. Both CCDCs are heavily intertwined within the COMMD-ring, but take different routes around the ring, with CCDC93 proceeding anticlockwise (4 and 2 in the front; 5, 9 and 6 in the back) and CCDC22 clockwise (1, 7 and 10 in the front; 5, 3 and 8 in the back; Fig. 3d,i) in an N- to C-terminal direction. Both CCDCs end up in the I-coil region where they form a heterodimeric coiled coil that extends into the bottom half of the structure.

CCDC93 resides next to the N-terminal domains (NTDs) of COMMD2 and COMMD4 (Fig. 3i,j). The NN–CH domain binds on the side of COMMD4 NTD, while α7 of CCDC93 binds an interface formed by α2–α3 loop, α4 and α7 of COMMD4. NTDs of several COMMDs (2, 3, 4, 5, 7 and 10) bind a segment of the CCDCs via this interface, which we call the peptide-binding site (Extended Data Fig. 3e). The α3–α4 loop of CCDC93 contacts the extended C-terminal helix (CH) of COMMD2 via electrostatics mediated by Asp67 and Asp69 (Fig. 3j). CCDC93 proceeds to circle around the NTD of COMMD2 in a headlock, binding the peptide-binding site via α9 and forming a long helix-loop-helix motif (HLH) (α10 and α11) that reaches all the way around to the NN–CH domain (Fig. 3j). The flexible loop region contains a phosphorylation site on Thr234 (Fig. 3j and Supplementary Data 2). The protein chain then returns toward the I-coil region of the COMMD-ring. The long loop in the HLH motif is flexible and therefore not visible in our cryo-EM map, but density features support our assignment of the two helices. The final helices (α12 and α13) predominantly bind the COMMD domain of COMMD7, with α12 binding in a tight hydrophobic pocket, before reaching the I-coil region (Fig. 3k).

CCDC22 occupies the lower half of the COMMD-ring. Its N-terminal NN–CH domain can readily be identified in the focused maps (Fig. 2c,d). The first helix of CCDC22 in the consensus map is the α8, which binds the COMMD domains of COMMD2, COMMD3 and COMMD8 at the N-terminal end of its CH as well as parts of the β-sheet (Extended Data Fig. 3c). It then binds the peptide-binding site on COMMD5 via a short α-helix containing motif (Fig. 3l and Extended Data Fig. 3e), before forming a bilateral binding interface between NTDs of COMMD3 and COMMD8 (Fig. 3l). At this point, a loop extends out of the COMMD-ring and is not visible in the density map, indicating flexibility (Fig. 3l). Finally, α14 of CCDC22 binds COMMD domains of COMMD1, COMMD6 and COMMD8 in a similar topology to α8, before reaching the peptide-binding sites on COMMD7 and COMMD10, and meeting CCDC93 at the I-coil region (Fig. 3i and Extended Data Fig. 3d).

### DENND10 and Retriever are scaffolded by CCDC22 and CCDC93

The CCDCs form a long heterodimeric coiled coil with three flexible corners and three major coil regions that we call the I-coil, R-coil and

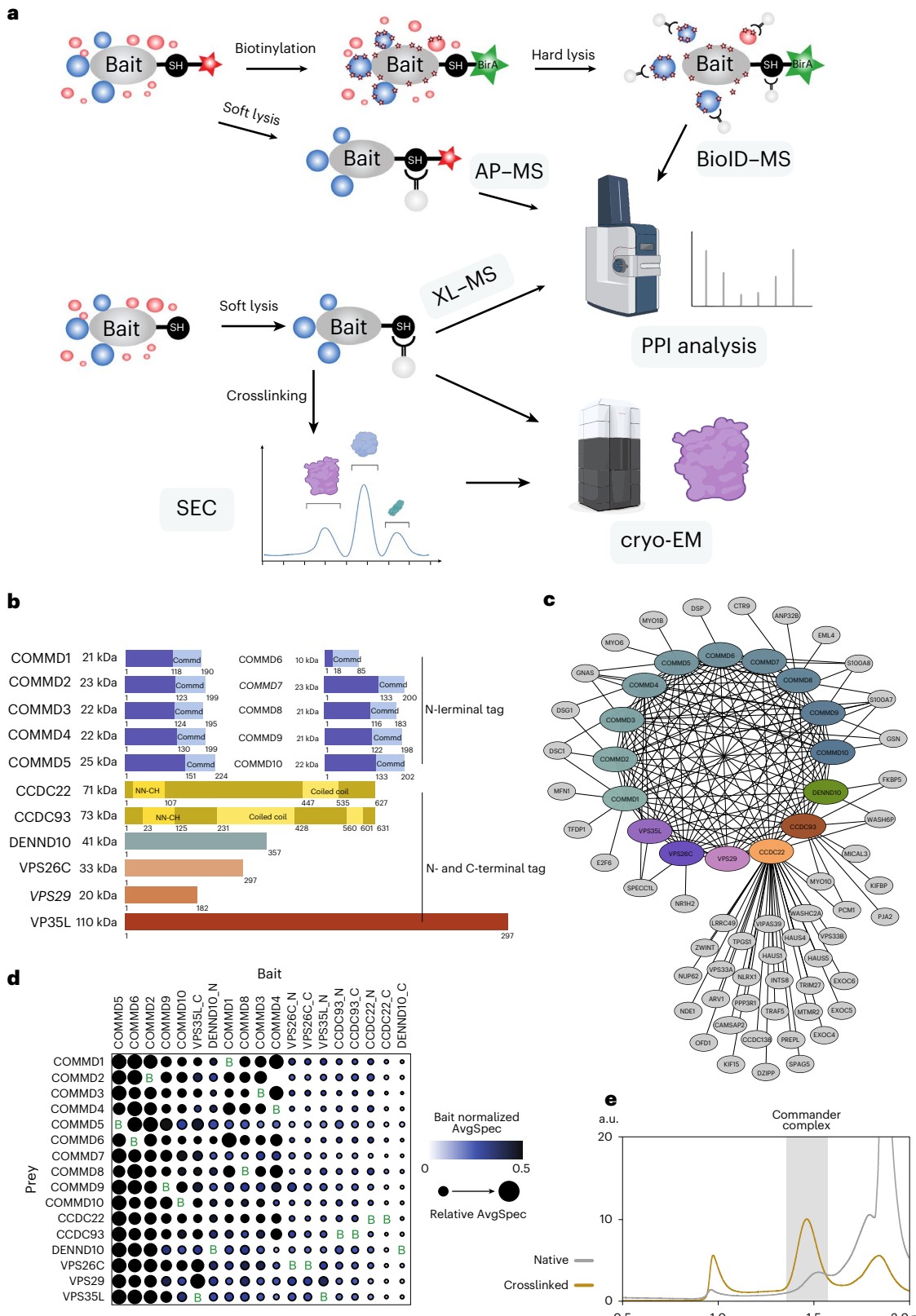

**Fig. 1 | Purification and analysis of the endogenous Commander complex.**
**a**, Schematic of the study design using AP–MS, proximity-dependent BioID–MS, crosslinking (XL)–MS, size-exclusion chromatography (SEC) and cryo-EM.
**b**, The known 16 members of the Commander complex proteins, their molecular weights (in kDa) and known domain compositions. The 14 complex proteins used as baits in the studies are shown in a normal, roman typeface. **c**, High-confidence and stable Commander complex interactome identified by AP–MS

analysis. **d**, Stoichiometry analysis of N- or C-terminally tagged Commander complex components identified with AP–MS. The color of each circle represents the abundance of each prey normalized to the mean abundance of the bait, and the circle radius indicates the relative abundance across all samples. **e**, Size-exclusion chromatography of the purified Commander complex with or without crosslinking. The peak indicated in gray background was used for cryo-EM analysis. a.u., arbitrary units.

**Table 1 | Cryo-EM data collection, refinement and validation statistics**

| | Native Commander complex (EMDB-17342) | Crosslinked Commander complex (EMDB-17340), (PDB 8POW) | Crosslinked Commander complex, focused map 1 (EMDB-17339), (PDB 8POV) | Crosslinked Commander complex, focused map 2 (EMDB-17341), (PDB 8POX) |
|---|---|---|---|---|
| **Data collection and processing** | | | | |
| Magnification | 165,000 | 105,000 | 105,000 | 105,000 |
| Voltage (kV) | 300 | 300 | 300 | 300 |
| Electron exposure (e⁻/Å²) | 42.8 | 59.0/56.0 | 59.0/56.0 | 59.0/56.0 |
| Defocus range (μm) | 1.5–3.0 | 0.4–2.2 | 0.4–2.2 | 0.4–2.2 |
| Pixel size (Å) | 0.820 | 0.846/0.862 | 0.846/0.862 | 0.846/0.862 |
| Symmetry imposed | C1 | C1 | C1 | C1 |
| Initial particle images (no.) | 137,000 | 5,372,000 | 667,000 | 667,000 |
| Final particle images (no.) | 90,000 | 667,000 | 125,000 | 13,000 |
| Map resolution (Å) | 3.3 | 2.9 | 6.5 | 7.5 |
| FSC threshold | 0.143 | 0.143 | 0.143 | 0.143 |
| Map resolution range (Å) | 3.3–44.2 | 2.9-40.0 | 6.5-40.0 | 7.5–40.0 |
| **Refinement** | | | | |
| Model resolution (Å) | | 3.03 | 7.94 | 8.14 |
| FSC threshold | | 0.5 | 0.5 | 0.5 |
| Model resolution range (Å) | | 3.03–40.0 | 7.94–40.0 | 0.814–40.0 |
| Map sharpening B factor (Å²) | | −77 | −546 | −511 |
| Model composition | | | | |
| Nonhydrogen atoms | | 17,518 | 9,073 | 1,850 |
| Protein residues | | 2,208 | 1,802 | 1,850 |
| Ligands | | 0 | 0 | 0 |
| *B* factors (Å²) | | | | |
| Protein | | 32.14 | 185.77 | 378.41 |
| Ligand | | | | |
| R.m.s. deviations | | | | |
| Bond lengths (Å) | | 0.004 | 0.006 | 0.009 |
| Bond angles (°) | | 0.644 | 1.199 | 1.409 |
| **Validation** | | | | |
| MolProbity score | | 0.97 | 1.48 | 1.79 |
| Clashscore | | 1.42 | 3.50 | 9.06 |
| Poor rotamers (%) | | 0.41 | 0 | 0 |
| Ramachandran plot | | | | |
| Favored (%) | | 97.57 | 95.07 | 95.64 |
| Allowed (%) | | 2.39 | 4.54 | 4.25 |
| Disallowed (%) | | 0 | 0.39 | 0 |

V-coil (Fig. 4a and Extended Data Fig. 4a,b). Two high-confidence crosslinking–mass spectrometry (XL–MS) crosslinks between Lys373 of CCDC22 and Lys352 of CCDC93 (I-coil) as well as Lys598 of CCDC22 and Lys601 of CCDC93 (V-coil) support our atomic model (Extended Data Fig. 4d and Supplementary Data 2). The I-coil begins close to the COMMD domains of COMMD1 and COMMD6. Supporting our assignment, one XL–MS crosslink was identified between Lys318 of CCDC93 and Lys114 of COMMD7 (Extended Data Fig. 4d and Supplementary Data 2). This crosslink is likely partially responsible for the reduction in conformational heterogeneity upon crosslinking. Furthermore, another crosslink between the NTDs of COMMD2 (Lys42) and COMMD10 (Lys56) supports our assignment of the COMMD-ring chains (Extended Data Fig. 4d and Supplementary Data 2).

The NN–CH domain of CCDC22 is wedged between the C-terminal V-coil of CCDC93/22. The C-terminal half of the V-coil provides a hydrophobic binding interface, while the N-terminal half provides a hydrophilic one. In the complete model, the Cα–Cα distance between the last residue of the CCDC22 NN–CH and the first residue of α8 is 23 Å, a distance that accommodates the 12 residues in the linker between them. This is the only covalent link on this side of the complex between the top and bottom halves, which highlights the importance of the binding interface between the CCDC22 NN–CH and the V-coil in terms of flexibility of the bottom half. The phosphorylation site detected at Ser54 is unlikely to regulate this binding as it is located away from these surfaces and is instead likely regulating external interactions (Fig. 4a and Extended Data Fig. 5a).

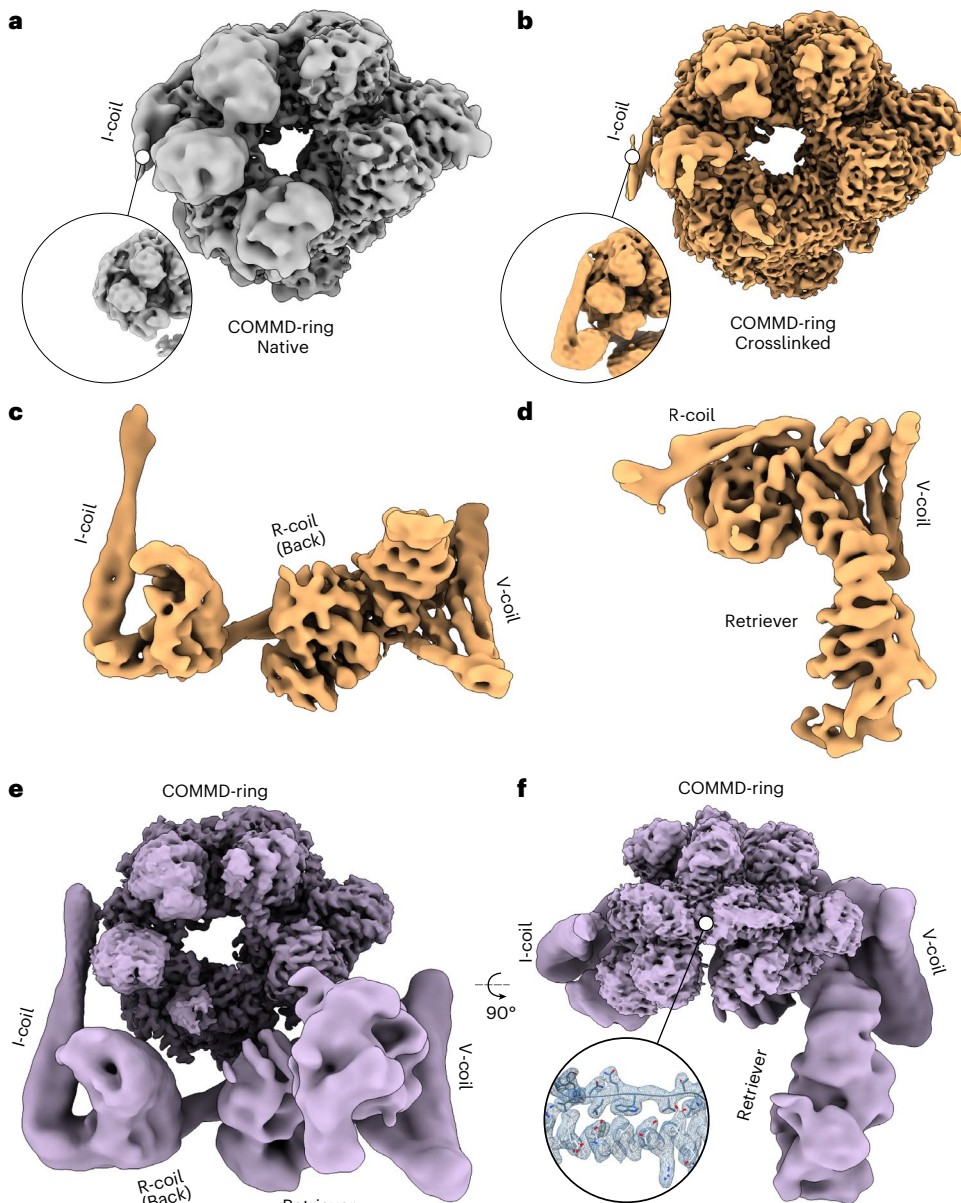

**Fig. 2 | Cryo-EM maps of the Commander complex. a,b**, Cryo-EM maps of the Commander complex COMMD-ring from native (gray) (**a**) and from crosslinked (gold) (**b**) samples. The insets show the I-coil region from both maps at lower isosurface threshold. **c**, Focused cryo-EM density map with DENND10 and CCDC22/93 coiled coils (focused map 1). **d**, Focused cryo-EM density map with Retriever subregion (focused map 2). **e**, A composite map of the complete Commander complex. **f**, Same as in **e**, rotated 90° around the *x* axis as indicated. The inset shows the structural detail of the map from within the COMMD-ring.

The I- and R-coils are oriented at a 62° angle with the N-terminal side of R-coil harboring the binding site for DENND10 (Figs. 3i and 4b). The binding site is composed of two interaction surfaces, with a hydrophobic groove between the CCDC93/22 coil and the corresponding hydrophobic patch on the NTD (or lobe) of DENND10 comprising the main site, and another mediated by electrostatic interactions via Glu19, Asp21, Glu25 and Arg40 of DENND10 (Fig. 4b). The putative Rab binding site of DENND10 (Extended Data Fig. 4e) is occupied by I-coil. The inherent flexibility of this region (Fig. 4g), however, may enable binding of Rabs in another conformation.

The AlphaFold2 (AF2)-predicted model for Retriever is analogous to that of Retromer, containing the distinctive α-solenoid fold at the center (Extended Data Fig. 4c). In our model of Retriever fit to our focused map (Fig. 4c), the α-solenoid is more compacted and bent at the VPS26-binding end than in Retromer (Extended Data Fig. 4f), shown by the decrease in distance of centers of mass of the ends

(109 Å in Retromer, 91 Å in Retriever). VPS35L has a long N-terminal extension of 180 residues, which is absent in VPS35. At the VPS26C binding end, the N-terminal extension contributes two additional helices (α2–α3). The tip extension (containing α3) provides a putative additional binding interface with VPS26C (Fig. 4d). We note that relative to the entire map, the cryo-EM density in this area of the map is the weakest, suggesting structural flexibility, and the AF2 prediction confidence is the lowest. In the AF2 prediction, the N-terminal extension continues toward the VPS29-binding end of VPS35L but is unstructured in the intervening region. We find several phosphorylation sites in this region in our AP–MS data, indicating that this site could be a hotspot for phosphoregulation (Fig. 4c and Extended Data Fig. 5a).

The very N-terminal part of the extension, the N-terminal tail, is predicted to contact VPS29 and reach all the way to the C-terminus of VPS35L. This is supported by the density in our reconstruction, and we

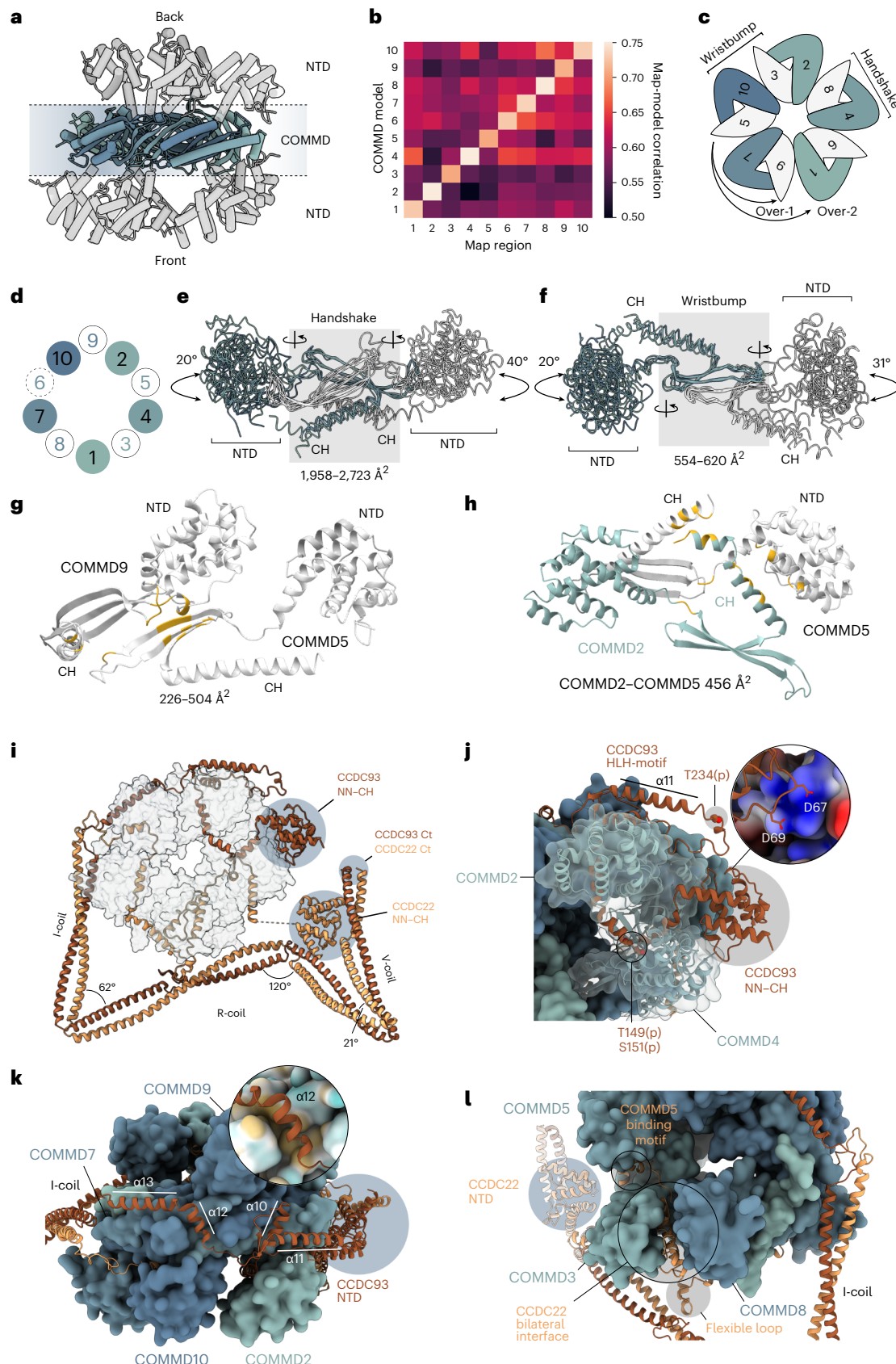

have therefore included the extension and the intervening unstructured region in our model (Fig. 4c). The N-terminal extension increases the binding interface between VPS35L and VPS29 substantially from 1,984 to 2,903 Å². Compared to Retromer, this binding interface is nearly twice as large, suggesting tighter binding in Retriever (Fig. 4e). Retriever itself binds CCDCs at the hinge between the R- and V-coil that forms a 120° angle, forming a large surface of mostly charged residues (Fig. 4c).

**Fig. 3 | Structure of the COMMD-ring. a**, The structure of the COMMD-ring is shown as a ribbon representation from the side. COMMD domains and NTDs are indicated. **b**, Heatmap depicting model-to-map cross-correlation coefficients for COMMD domains placed in each COMMD map region after real space refinement. **c**, Schematic diagram of COMMD-ring subunit organization. COMMDs with NTDs toward the viewer are indicated in blue-green shades, while COMMDs with NTDs away from the viewer are indicated in white. Handshake, wristbump, over-1 and over-2 interactions are indicated. **d**, Schematic representation of the NTD organization within the COMMD-ring. Numbers indicate COMMD proteins. Colors as in **c**. COMMD6 is indicated in dashed lines as it lacks an NTD. **e,f**, Superposed COMMD chains depicting the handshake (**e**) and wristbump (**f**) interactions between two chains with range of interaction surface areas for each COMMD pair. Angles of NTD rotation along the indicated hinge axes are shown. Colors as in **c**. **g**, Model of over-1 interaction between COMMDs 2

and 5. Interacting residues in gold, with buried surface area ranges indicated. **h**, Model of over-2 interaction between COMMDs 5 and 2, colors as in **g**. **i**, Overview of CCDCs intertwined within the COMMD-ring. **j–l**, Major CCDC interaction sites with COMMD chains. **j**, CCDC93 NN–CH domain binds the side of COMMD4 NTD and encircles the COMMD2 NTD in a headlock with HLH motif contacting the NN–CH domain. Experimentally identified phosphorylation sites are indicated. **k**, Top view of the COMMD-ring, showing the arrangement of CCDC93 α10–α13. The inset shows the hydrophobic binding pocket of α12 on COMMD7. Coloring indicates hydrophobic (yellow) and hydrophilic (cyan) surfaces. **l**, CCDC22 forms a bilateral interaction interface between COMMD3 and COMMD8. The loop protruding out of the density map between α13 and α14 of CCDC22 as well as the NN–CH domain of CCDC22 are indicated. NTD, N-terminal domain; CH, C-terminal helix; Ct, C-terminus.

---

Finally, we computationally analyzed structural flexibility within the complex. We found that despite seemingly not interacting with DENND10 or VPS29/VPS35L-N-terminal tail in the consensus cryo-EM map, COMMD1 NTD displays at least two alternative conformations where these interactions are imminent (Fig. 4f). This could indicate that the COMMD1 NTD has an important role in mediating interactions between Retriever and DENND10. The flexibility of the bottom half is facilitated by two main chain contact sites at each end, namely via the I-coil and the linker between the NN–CH of CCDC22 and its top half portion. Other flexible joints are also present, including the tethers between the I- and R-coils as well as the R- and V-coils. However, these are likely stabilized by binding of DENND10 and Retriever, respectively.

### Commander complex is a compact assembly

Recently, Healy et al. published an integrated structural model for the Commander complex[57]. As the model differs strikingly in overall arrangement from the structure presented in this study (Extended Data Fig. 4g), we compared the structures first by superposing the Healy et al. model using different alignment centers to our model (Fig. 5). The CCDC scaffolding is similar in both complexes, and the COMMD-ring, DENND10 and the Retriever subcomplex are in similar positions along it (Extended Data Fig. 4g). Three main differences between the models were found (Fig. 5). (1) The overall structure is more compactly packed in our model than in the Healy model. (2) The orientation of the COMMD-ring relative to the CCDC scaffolding is different so that in the Healy model the COMMD-ring lacks contact to the I-coil, which is evident both in our cryo-EM and XL–MS data (Fig. 2a,b and Extended Data Fig. 4d). Notably, the Healy model is incompatible with the cross-link between CCDC93 and COMMD7 detected in this study. Furthermore, the relative orientation between COMMD-ring and DENND10 or V-coil differ by 76° and 117°, respectively. (3) The twistedness of the scaffolding is dissimilar such that the relative orientation of DENND10 and Retriever differs by 65°.

Healy et al.[57] compiled the overall model of the Commander complex using AF2 combined with X-ray crystallography and cryo-EM data from certain regions of the complex. When compared to our structure, models of the COMMD-ring align well as both studies base

the structural models on high-resolution cryo-EM data (Extended Data Fig. 6b). CCDC22 helices α15 and α16, and the HLH motif of CCDC93 are placed differently in the two models. They are absent in the cryo-EM structure by Healy et al. (Electron Microscopy Data Bank, EMDB-28827, Protein Data Bank (PDB) ID 8F2U) whereas their placement in our model is supported by cryo-EM density (Extended Data Fig. 4h). The conformation of DENND10, I-coil and R-coil is based heavily on AF2 prediction in both models, as our cryo-EM reconstruction has limited resolution in this region (Extended Data Fig. 5d), and the Healy model is entirely based on AF2 prediction for this part (Extended Data Fig. 6c). The overall folds are similar, except that an interaction between the N-lobe of DENND10 and I-coil presented in the Healy model (indicated with an asterisk, Extended Data Fig. 6c) is not featured in our model. This may be explained by conformational heterogeneity, as evidenced by our three-dimensional (3D) variability analysis of this region (Fig. 4f).

In our model, the Retriever subcomplex extends out of the main body of the complex (Fig. 2e,f). The tip extension of VPS35L binds VPS26C, while in the Healy model it forms an interaction surface with the CCDC22 part of V-coil (Extended Data Fig. 6d). This interaction seen in the Healy model was predicted by AF2 and may reflect conformational heterogeneity in this region. Finally, Healy et al.[57] solved the crystal structure of VPS29 with VPS35L (24–38) peptide (PDB ID 8ESE), which is consistent with our cryo-EM structure (Extended Data Fig. 6e).

### The interactome of Commander extends into cell cycle pathways

To provide further insight into the molecular context, cellular interactions and functions of the Commander complex, we focused on the transient HCIs identified through BioID–MS analysis (Fig. 1 and Extended Data Fig. 7a). In total, the BioID analysis resulted in the identification of 148 interacting proteins corresponding to 500 HCIs for the 19 Commander complex protein constructs used as baits. We compared our HCIs to previously reported interactions in six public interactome databases and observed that most of the interactions we identified were previously unreported (Fig. 6a and Supplementary Data 1). Our approach, therefore, provides a comprehensive view of

---

**Fig. 4 | Structures and interactions of the effector subunits. a**, CCDC22/93 coiled-coil regions, NTD of CCDC22 and DENND10 in density. The inset shows the binding site of CCDC22 NN–CH between the C-terminal V-coil characterized by hydrophobic and hydrophilic interfaces. Ser54 phosphorylation site identified in this study is indicated. **b**, DENND10 interface with R-coil of CCDC22/93. The major interface consists of a hydrophobic core along the groove of the R-coil and a charged patch on the C-terminal side of R-coil. **c**, The structure of the Retriever subcomplex in the context of Commander. Four experimentally identified phosphorylation sites (Ser70, Ser71, Thr76 and Ser80) are indicated in the disordered intervening N-terminal region of the model. The insets show the Retriever interfaces with CCDC22/93 through the C-terminal part of VPS35L, mainly via charged interactions. The N-terminal extension of VPS35L is shown in

density where contributions from VPS29 and VPS35L C-terminal parts have been masked and low-pass filtered. **d**, Interfaces between the mobile NTD of COMMD1 with both DENND10 and VPS29/VPS35L. The insets show the first, middle and last frames of component 1 from 3D variability analysis (3DVA), reconstructed from particles using the intermediates job type. **e**, Interaction between VPS35L and VPS29 (top) compared to corresponding interaction in fungal Retromer arch (PDB 6H7W). The N-terminal extension of VPS35L accounts for nearly 33% of the buried interface. **f**, Interaction between VPS35L and VPS26C compared to the fungal VPS35–VPS26 interface. The insets show that VPS35L contains an extended helix that provides an expanded binding interface to VPS26C. Side chains in **a** and **b** are included only for visualization purposes.

---

the Commander complex interactions and sheds light on its cellular role. As in our AP–MS data, a large portion of the detected transient HCIs is specific for C-terminally tagged CCDC22.

We investigated the molecular interactions formed by the Commander complex in greater detail by creating a comprehensive map that integrated existing protein complex information (CORUM), pathway

information (Reactome Pathway database) (Extended Data Fig. 7c), and biological processes (from Gene Ontology) for the Commander complex interactions identified using both AP–MS (red edges and/or lines) and BioID–MS (blue edges and/or lines; Fig. 6a). The analysis revealed several known complexes that were overrepresented in our data, including the WASH complex, the distal appendage, HAUS

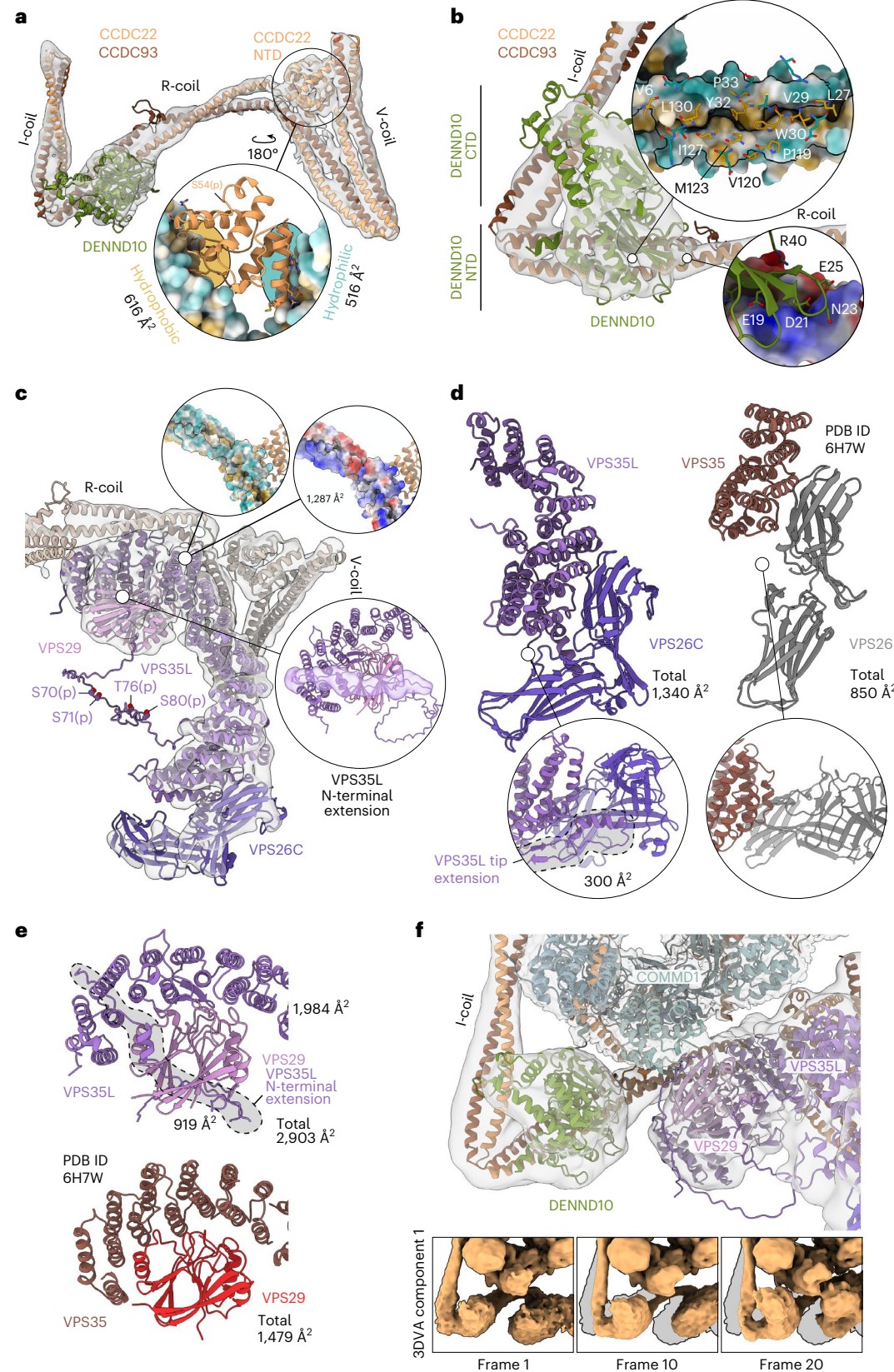

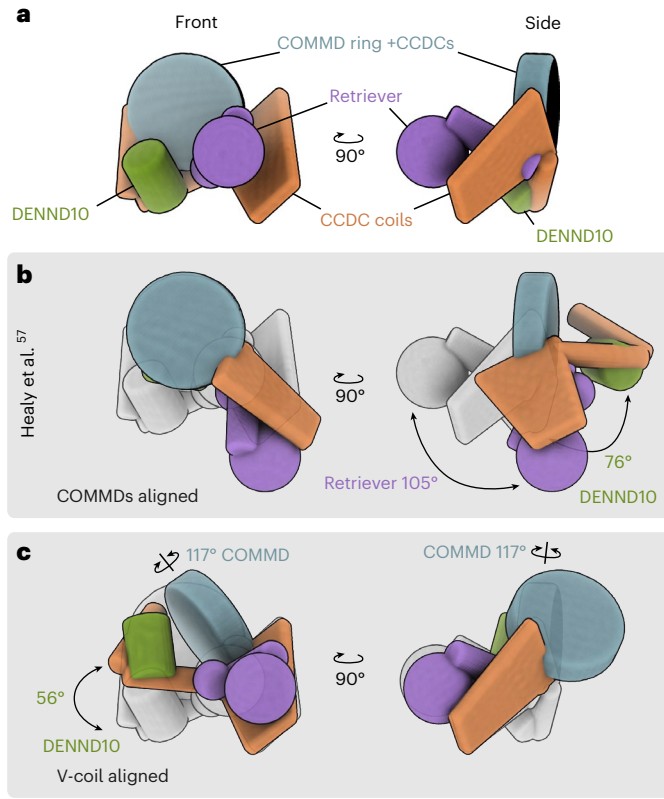

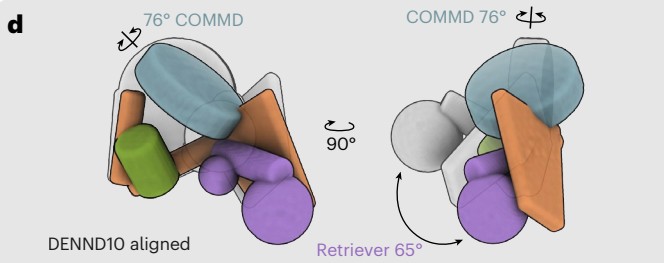

**Fig. 5 | Analysis of overall tertiary fold of the endogenous Commander complex compared to existing literature. a**, A simplified 'ragdoll' representation of major components of the Commander complex from this study. **b–d**, The overall structural model from Healy et al.[57] highlight the major differences between these models with alignment centers of the models located at the COMMD-ring (**b**), V-coil (**c**) and DENND10 (**d**). The COMMD-ring is represented by a disc aligned to the COMMD domains, the coiled-coil domains are represented by cylinders (I- and R-coils) or a trapezoidal prism (V-coil + CCDC22 NN–CH), DENND10 as a cylinder and Retriever subcomplex as spheres (VPS29, VPS26C + N-terminal half of VPS35L α-solenoid) or a cylinder (VPS35L C-terminal half of VPS35L α-solenoid). Component relative rotation angles are calculated based on the underlying atomic coordinates of backbone Cα atoms.

complex, BORC complex, Exocyst complex and the TPGC (tubulin polyglutamylase complex). Additionally, we identified several other clusters, such as the CCDC protein cluster, cell cycle and/or division related proteins and transport and/or cargo proteins.

Transient interactions of the Commander proteins are dominated by intra-complex interactions. Whereas in the AP–MS interactome the entire Commander complex was detected with all baits, structure-dependent variation can be seen in the transient BioID HCIs (Extended Data Fig. 7a). BirA*-tagged DENND10, CCDC22 and CCDC93 biotinylate each other as well as both the COMMD proteins and the Retriever subcomplex. However, due to the strong distance dependence of the BioID signal, VPS26C detects only the Retriever subcomplex and CCDC93 but none of the COMMD proteins that reside further away. We subsequently analyzed bait–bait clustering of BioID

interactions of the Commander proteins to infer structural regions with similar interaction profiles (Fig. 6b). We found two distinct clusters of binding partners, suggesting that the Commander complex exhibits two protein-binding regions. These clusters are similar to the clusters identified by AP–MS but follow a neat division along WB interfaces in the complex structure mapped in this study (Fig. 6b, upper and lower insets). This indicates that the BioID data can provide information on which side of the Commander complex the other interacting complexes would reside. The 'MS microscopy' analysis of transient HCIs showed a predominant localization for the complex proteins to Golgi in agreement with Human Protein Atlas database but contrasting previous studies that localized it predominantly to endosomes[20,33,57–59]. Several components also displayed clear microtubule (COMMD1–4, -8) and centrosomal (CCDC93 and CCDC22) localization (Extended Data Fig. 7d).

We next performed Gene Ontology Cellular Component (GO-CC) and Biological Process (GO-BP) term analysis and clustering based on the identified BioID interactions (Fig. 6c,d). The GO-CC term 'WASH complex' was highly (>6-fold) enriched with several of the complex components. COMMD3, 5, 8, 9 and 10, DENND10, VPS26C, VPS35L, CCDC22 and CCDC93 form transient HCIs with WASH complex components WASHC2A, WASHC2C, WASHC3, WASHC4 and WASH1C protein homolog WASH3P (Fig. 6e). WASH complex regulates actin nucleation, which is supported by VPS26C interactor F-actin capping protein CAPZA3. All analyzed complex components associate with endosome and centrosome related CC-terms. This is supported by COMMD8 and CCDC22 interaction with adaptor protein AP4E1 that functions in cargo selection[60]. Further, VPS35L interacts with trafficking proteins TMED5, UBL4A and HERC1, while DENND10 and CCDC22 interact with the endosomal transport protein FKBP15. Additionally, CCDC22 had HCIs to several transport or cargo proteins, including lysosomal transporter BORC complex[61] (Fig. 6c). As CCDC22 has a large portion of total HCIs, a total of seven cilia and microtubule associated CC-terms are highly enriched from its interactions (Fig. 6c). These proteins are shown as part of a large 'Distal appendage' cluster in Fig. 6e. Additionally, CCDC22 showed high (>6-fold) enrichment for other important complexes: the procentriole replication complex, NatC complex, BORC complex and HAUS complex (Fig. 6c,e).

The Commander complex is involved in diverse cellular biological processes, including five related to trafficking and one to immune response (Fig. 6d). Excluding CCDC22, the interactome is dominated by proteins of Commander and WASH complexes, in agreement with the earlier proposed roles of the Commander complex. COMMD1, -2, -4 and -6 lack these terms, likely due to their location in the complex away from the WASH complex binding interface. COMMD1, -2 and -9, interact with E2F6 and COMMD8 with E2F4, both cell cycle regulating transcription factors. Further, COMMD1 and COMMD8 interact with E2F regulatory proteins TFDP2 and TFDP1, respectively. Another connection to transcription regulation comes from COMMD10 that interacts with the membrane-bound NF-kB regulator TRIM13 (Fig. 6e). Intracellular transport regulation, centriole replication, centrosomal targeting and cilium assembly were significantly enriched for CCDC22, VPS35L and CCDC93. This group of proteins contain proteins related to the distal appendage, and a large group of other centrosomal and ciliary proteins, containing OFD1, PCM1 and MTMR2 that were also detected in AP–MS (Fig. 6e). Most of the Commander proteins interact with the cilia modifying TPGC[62]: COMMD1, -2, -9 and -10, VPS26C, VPS35L and DENND10 interact with TPGS1, COMMD2 and -9, VPS26C and VPS35L interact with LLRC49, and CCDC22 with TTLL5. Additionally, CCDC22 had moderate enrichment (>3-fold) of large number of additional GO-BP terms involved in processes related to centrosomes, microtubules and intracellular transport and to processes related to membrane-bound organelles (Fig. 6d and Supplementary Data 2). Although CCDC22 is an integral part of the Commander complex, we cannot rule out the possibility that some of these functions are Commander independent.

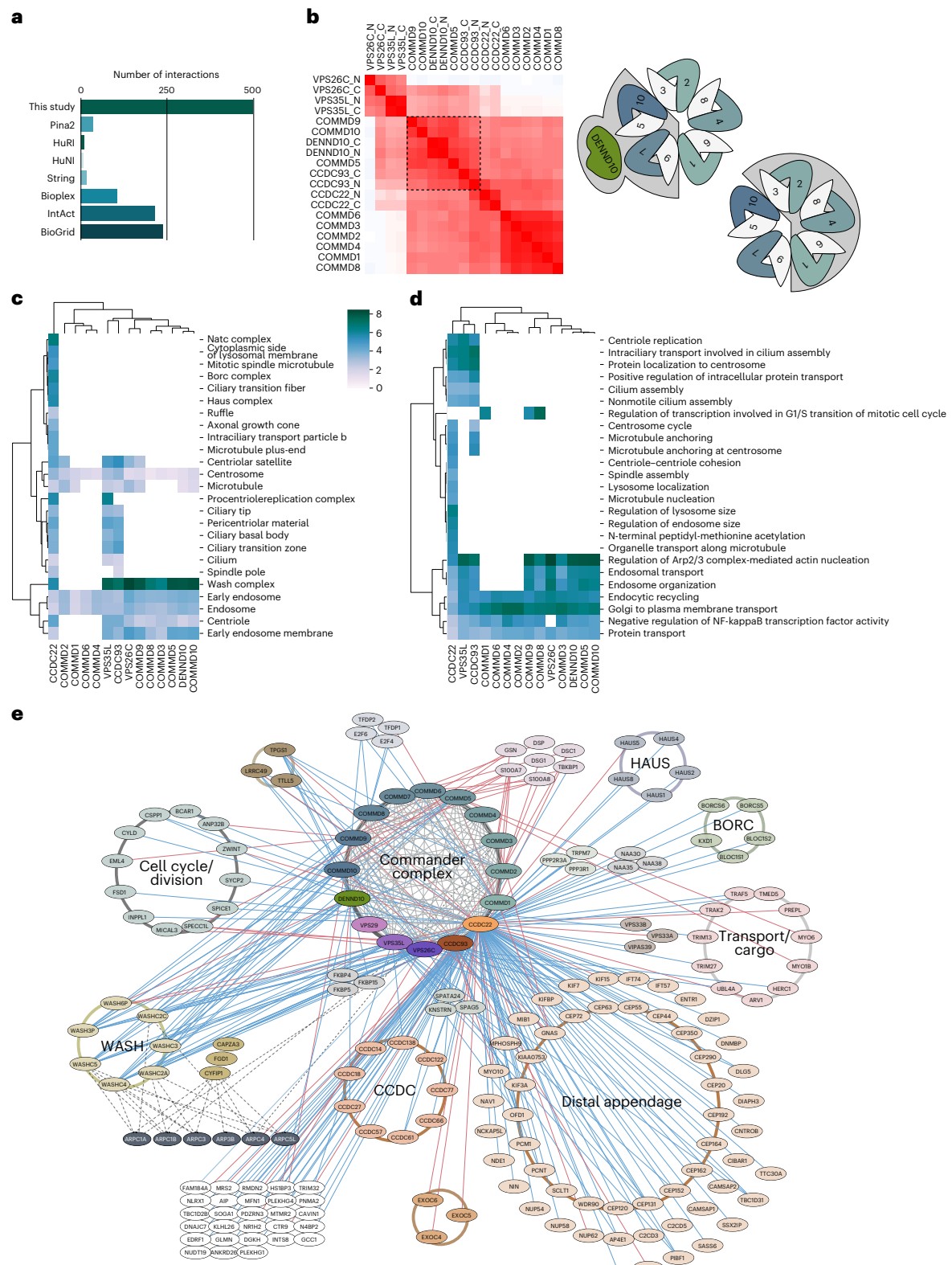

**Fig. 6 | Molecular context, cellular interactions and functions of the Commander complex. a**, Comparison of high-confidence Commander complex interactions (from BioID–MS) to interactions in databases. **b**, Bait–bait clustering of BioID interactions of Commander proteins reveals two distinct clusters suggesting two different subcomplexes. **c**,**d**, GO-CC (**c**) and GO-BP (**d**) term analysis and clustering based on Commander complex BioID interactions. **e**, Complete map of the molecular interactions formed by the Commander complex. The key shows the Commander complex components used as baits color coded as in Fig. 1, identified interactions with AP–MS and BioID are shown as red and blue edges, respectively. Reciprocal interactome analysis with ARP proteins is shown in the lower right corner. Interacting proteins are organized to designated protein complexes (CORUM; with brown to light orange-colored circles), and based on their function (light gray circles). The nodes linked to cilium assembly (based on GO-BP) are shown with a light orange node color.

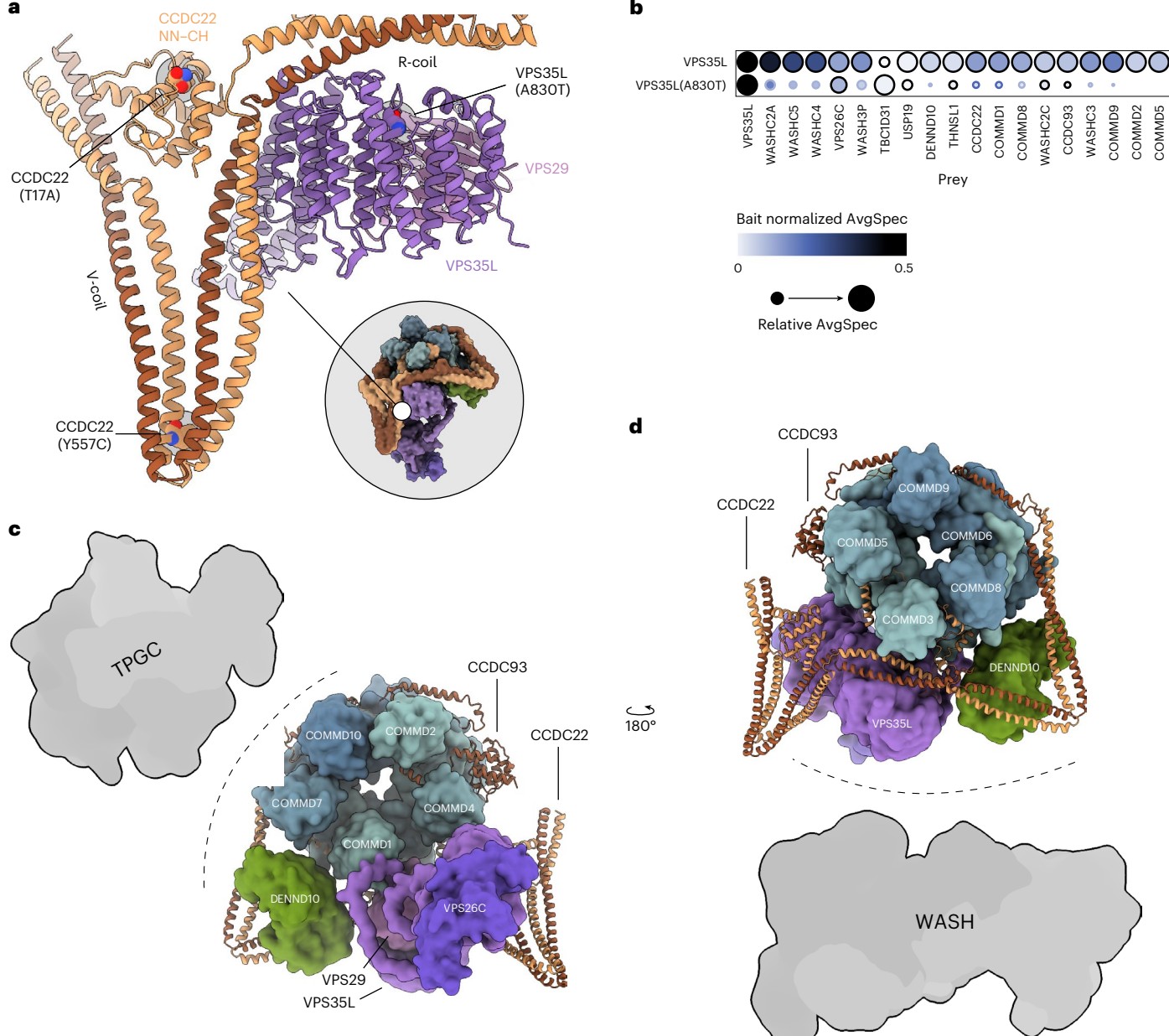

**Fig. 7 | RSS and XLID related mutations and putative interaction interfaces of the Commander complex. a**, Three mutations associated with RSS or XLID highlighted within the context of the Commander complex structure. **b**, Effect of A830T mutation on VPS35L in BioID–MS. All PPIs passing HCI criteria to either wild-type VPS35L or VPS35L(A830T) are plotted for both constructs, with HCIs indicated using a black outline and non-HCIs with a light blue outline. Node color corresponds to the bait normalized abundance of the average spectral count for each prey, and node radius to its relative abundance across all baits determined by ProHits-Viz. **c**, Composite model of the Commander complex, indicating putative interaction interfaces with TPGC. **d**, Rotated view of the model in **c**, with putative interaction interface of the WASH complex indicated.

As expected, interactions with the WASH complex were particularly prominent with many of the Commander complex proteins. To further explore this interaction, we re-examined the reciprocal analyzes we have performed earlier for the ARPC1A-B, ARPC3, ARP3B, ARPC4 and ARPC5L (ref. 36). With these baits, we could detect several of the WASH complex components from the opposite direction, as well as a previously unreported interaction of ARPC1B with COMMD2 (Fig. 6d).

**RSS-linked point mutations alter PPIs of CCDC22 and VPS35L**
The Commander complex has been associated with RSS or X-linked intellectual disability (XLID) via point mutations in VPS35L and CCDC22 (refs. 13,31,63,64). Disease variants CCDC22(T17A), CCDC22(Y557C) and VPS35L(A830T) (Figs. 2d and 7a) are listed as 'pathogenic' for RSS in

GnomAD database[65]. Cell lines expressing these disease variants were generated for BioID–MS analysis to investigate their effect at the PPI level (Supplementary Data 1).

The VPS35L(A830T) has been suggested to abolish its interaction with VPS29 (ref. 63). Our data shows that the A830T mutation does not inhibit interaction with VPS26C but separates VPS35L from the rest of the Commander complex and disrupts its interaction with the WASH complex (Fig. 7a,b).

With the CCDC22 RSS disease variants, we discovered fewer interactions with COMMD proteins (especially to COMMDs -3, -6 and -7), whereas interactions with the WASH complex become pronounced (Extended Data Fig. 7b). Using the Reactome pathway analysis on the CCDC22 disease mutant interactors, we could detect enrichment of

the 'Cilium assembly pathway' for CCDC22(T17A) and 'Retrograde transport at the trans-Golgi network' for both variants (Extended Data Fig. 7e). The CCDC22(Y557C) mutation is situated at the tip of the CCDC22 part of V-coil, a region predicted to interact with VPS35L by Healy et al.[57]. Our BioID data show no major changes in the interactome that could be expected if this interaction was important. On the other hand, the distal location of VPS26C, and thus the tip of VPS35L, from the V-coil is supported by our BioID data where VPS26C and CCDCs are not in close proximity (Fig. 6b). However, possible effects on the VPS35L-V-coil interaction caused by this mutation need to be experimentally interrogated.

## Discussion

In this study, we determined the structure (Supplementary Video 1) and analyzed the interactome of the endogenous human Commander complex through a combination of cryo-EM and MS-based proteomics. Individual Commander components are involved in cellular functions in a wide variety of disease states and are expressed in all tissue types[66]. Therefore, malfunctioning individual components can exhibit pleiotropic effects. However, the molecular function of the complex remains unknown. Here, we report a set of biological processes in which Commander components are involved. We hypothesize that Commander acts as a facility operator that mediates crucial cellular processes from transcriptional regulation up to vesicle exocytosis and microtubule-based cell remodeling.

Our analyzes reveal that the Commander complex is organized into two distinct halves, each responsible for a unique set of functions. The bottom half contains two main effectors connecting the complex to COMMD proteins in the top half. The top half may serve as a recognition site or a platform for protein complex assembly via COMMD NTD binding sites. CCDC scaffold keeps the two halves together in one tightly packed unit where correctly folded CCDCs are essential for correct positioning of the COMMD-ring, DENND10 and Retriever within the complex. This structural organization provides the basis for the diverse interactions and roles of the Commander complex (Figs. 6 and 7).

Combined with the structure, systematic proximity labeling allows identification of interaction surfaces in medium to high molecular weight complexes. We found that the WASH complex was highly enriched with several of the Commander complex components: most BioID signals for WASH come from CCDC22, CCDC93, DENND10, VPS26C and VPS35L, which indicates that the binding interface is located between DENND10 and Retriever sides of Commander (Fig. 7c,d). This is in line with a previous report showing that WASH complex interacts with Retromer via VPS35 (ref. 67). Aside from its effect on Commander complex assembly, the RSS variant VPS35L(A830T) also displays reduced affinity for the WASH complex. The mutation site is located at the interface between VPS35L and VPS29, and therefore this observation fits well with our proposed WASH binding interface model where VPS29 is located at the center. By contrast, while CCDC22 variants (T17A and Y557C) are also near the putative WASH binding site, they have the opposite effect on WASH complex interactions, implying a complex molecular etiology of RSS.

We also identified an interface comprising COMMD1, COMMD8, COMMD9, COMMD10 and DENND10, exhibiting BioID signals for three TPGS1 subunits (TPGS1, LRRC49 and TTLL5). Together with COMMD7, these proteins form an interaction surface around the I-coil region of the Commander complex (Fig. 7).

The top half of the Commander complex contains flexible components (NTDs of COMMDs -1, -7, -9 and -10; HLH motif of CCDC93), and the bottom half exhibits compositional and conformational heterogeneity, particularly within the Retriever subcomplex. This may reflect physiological assembly and/or the function of the complex. Indeed, COMMD proteins and the CCDCs have been proposed to form a complex without Retriever, termed the CCC complex[68]. Assuming a similar head-to-head dimerization mode as Retromer, the structure

of Commander permits binding of the CCC complex on a Retriever dimer without obvious steric clashes (Extended Data Fig. 5f–h). Such analysis relies on having access to complete native structure featuring external surfaces and exposed domains facilitating biological functions in the cellular context. This highlights the need for experimental data in addition to in silico predictions for quaternary structure analysis of large macromolecular complexes (Fig. 5 and Extended Data Fig. 6).

The exact role of COMMD protein NTDs is unknown, but they may have a recognition function. NTDs of COMMDs share a structural fold but are less conserved than COMMD domains. The peptide-binding site is a potential candidate for linear peptide epitope interactions. The flexibility of the NTDs correlates with the occupancy of this binding site as empty sites on COMMD1 and COMMD9 are the most flexible, followed closely by COMMD7 and COMMD10, each binding a short helix between long flexible regions of CCDC22.

The role of DENND10 in the complex is also unknown. In the present structure, it is stably bound on the R-coil, but connection to the I-coil is unclear. Although the binding site identified in the homologous DENND1B with Rab35 (Extended Data Fig. 4f) is blocked in our consensus structure, the flexibility exhibited by the C-terminal half of DENND10 may expose this binding site (Fig. 4f)[57]. As COMMD1 NTD seems to move together with the DENND10, it is tempting to speculate that COMMD1 may regulate the availability of this site through allostery. Another possible mode of regulation comes from phosphorylation (Extended Data Fig. 5a). Four particularly interesting phosphorylation sites in the disordered intervening N-terminal region of VPS35L coincide with flexible regions in our model, which may indicate binding sites for interactors (Extended Data Fig. 5a). Perhaps the clearest site for direct binding regulation is the site on CCDC22, which is located on the only external face of the domain.

Our study provides a comprehensive characterization of the endogenous human Commander complex, including its structural architecture and cellular interactions. Our findings shed light on the molecular mechanisms pertinent to the functions of this complex. We validate its role in regulating intracellular trafficking and cell homeostasis, but we also discover associations with cilium assembly as well as centrosome and centriole functions. These findings pave the way for further research into the Commander complex, underlying causes of related diseases, in addition to potential drug discovery efforts targeting its components.

## Online content

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

## Methods

### Cloning of Commander complex components

A total of 14 human Commander complex components were obtained from the human ORFeome Libraries (Genome Biology Unit, HiLIFE (University of Helsinki), Horizon Discovery (Perkin-Elmer)). To generate stable isogenic and tetracycline-inducible cell lines, gene constructs were cloned using Gateway cloning, into N-terminal pTO_HA_StrepIII-N_GW_FRT and N- or C-terminal MAC-tagged vectors (BirA* enzyme for biotinylation). After verification by sequencing the constructs were subsequently introduced into Flp-In T-REx 293 cells (Life Technologies) as described by Liu et al.[36,69].

### Generation of RSS disease variants

RSS disease point mutations were introduced to CCDC22 and VPS35L genes by site-directed mutagenesis using Q5 High-Fidelity DNA Polymerase (NEB, catalog no. M0491) and the following primers: 5′-GGCGCGGCAGTTCCT and 3′-AACTGCCGCGCCGGC (CCDC22(T17A)), 5′-AAGGCCTGTAAGTATCTAGCTGC and 3′-GATACTTACAGGCCTTC CGAACA (CCDC22(Y557C)), and 5′-TCCACCATGAGCCAGGAG and 3′-CTCATGGTGGAGAGGAGATGC (VPS35L_A830). The disease variants and wild-type genes were cloned into C-terminal ultraID containing MAC3-tag vector[70].

### Cell culture

Human embryonic kidney 293 (HEK293) cell derivative Flp-In 293 T-Rex (Invitrogen, catalog no. R78007) was used to generate stable cell lines with a single copy of tetracycline-inducible, Strep-HA tagged transgene in their genome. The cells were cultured in low glucose tetracycline-free DMEM (Sigma Aldrich) supplemented with 10% FBS and 100 µg ml$^{-1}$ penicillin–streptomycin (Life Technologies) at 37 °C with 5% $CO_2$. For approximately $7 \times 10^7$ HEK cells stably expressing the human Commander complex components, protein expression was induced with 2 µg ml$^{-1}$ tetracycline for 24 h (AP–MS and BioID). An additional 50 µM of biotin was added for proximity labeling (BioID) for 24 h (MAC-tagged constructs) or 5 h (MAC3-tagged constructs). The cells were pelleted using centrifugation, snap frozen in liquid nitrogen and stored at −80 °C.

### Affinity purification

Cell pellets were suspended into 3 ml of lysis buffer (50 mM HEPES pH 8.0, 5 mM EDTA, 150 mM NaCl, 50 mM NaF, 0.5% IGEPAL, 1 mM DTT, 1.5 mM $Na_3VO_4$, 1 mM PMSF, 1× protease inhibitor cocktail, Sigma) and lysed on ice for 15 min. BioID lysis buffer was completed with 0.1% SDS and 80 U ml$^{-1}$ benzonase nuclease (Santa Cruz Biotechnology), and lysis was followed by three cycles of water bath sonication (3 min) with intervening resting periods (5 min) on ice. All samples were then cleared by centrifugation, and the supernatants were poured into microspin columns (Bio–Rad) that were preloaded with 200 µl of Strep-Tactin beads (IBA GmbH) and allowed to drain under gravity. The beads were washed three times with 1 ml of lysis buffer (without SDS for BioID samples) and then four times with 1 ml of lysis buffer without detergents and inhibitors (wash buffer). The purified proteins were eluted from the beads with 600 µl of wash buffer containing 0.5 mM biotin. To reduce and alkylate the cysteine bonds, the proteins were treated with a final concentration of 5 mM tris(2-carboxyethyl) phosphine and 10 mM iodoacetamide, respectively. Finally, the proteins were digested into tryptic peptides by incubation with 1 µg of sequencing grade trypsin (Promega) overnight at 37 °C. The digested peptides were purified using C-18 microspin columns (The Nest Group Inc.) as instructed by the manufacturer. For the mass spectrometry analysis, the vacuum-dried samples were dissolved in buffer A (1% acetonitrile and 0.1% trifluoroacetic acid in mass spectrometry-grade water).

### Mass spectrometry analysis and database searches

The samples were analyzed using the Evosep One liquid chromatography system coupled to a hybrid trapped ion mobility quadrupole time-of-flight mass spectrometer (Bruker timsTOF Pro) via a CaptiveSpray nano-electrospray ion source. An 8 cm × 150 µm column with 1.5 µm C18 beads (EV1109, Evosep) was used for peptide separation with the 60 samples per day methods (buffer A, 0.1% formic acid in water; buffer B, 0.1% formic acid in acetonitrile). The mass spectrometry analysis was performed in the positive-ion mode using data-dependent acquisition in PASEF mode with ten PASEF scans per topN acquisition cycle. Raw data (.d) acquired in PASEF[71] mode were processed with MSFragger[72] against the human protein database extracted from UniProtKB. Both instrument and label-free quantification parameters were left to default settings.

### Quantification and statistical analysis

Significance Analysis of INTeractome (SAINT) express v.3.6.0 (ref. [73]) and Contaminant Repository for Affinity Purification (CRAPome, http://www.crapome.org) were used as statistical tools for identification of specific HCIs from AP–MS and BioID data. Seventeen control runs with MAC-tagged GFPs were used as controls for SAINT analysis. Identifications with a SAINT-assigned Bayesian false discovery rate (BFDR) ≥0.05 were dropped, as well as any proteins that were detected in ≥20% of CRAPome experiments, unless the spectral count fold change was >3 when compared to CRAPome average. The remaining HCIs were then used for further analysis. Interactomes were visualized using Cytoscape v.3.8.2 (ref. [74]) and ProHits-Viz[75].

### Databases to map known interactions

Known interactors were mapped from BioGRID (only experimentally detected interactions)[76], Bioplex (interactions with a probability >0.95)[77], human cell map[59], IntAct (only experimentally validated physical interactions)[78], PINA2 (ref. [79]) and STRING (only with a STRING score greater >0.9) databases[80]. Domain annotations were mapped from PFam[81]. Reactome annotations from UniProt to the lowest pathway level mapping file available at Reactome[82]. Gene Ontology and CORUM[83] annotations were taken from UniProt. GO-CC annotations for CORUM complexes were taken from the CORUM database[83].

### Cryo-EM sample preparation

For the cryo-EM analyses, $2 \times 10^9$ Flp-In T-REx 293 cells stably expressing the N-terminally Strep-tagged human COMMD9 protein were induced with 2 µg ml$^{-1}$ tetracycline for 24 h. The cells were pelleted using centrifugation, snap frozen in liquid nitrogen and stored at −80 °C. The sample was then suspended in 80 ml of lysis buffer (50 mM HEPES pH 8.0, 5 mM EDTA, 150 mM NaCl, 50 mM NaF, 0.5% IGEPAL, 1 mM DTT, 1.5 mM $Na_3VO_4$, 1 mM PMSF and 1× protease inhibitor cocktail (Sigma)) on ice. The sample was then cleared by centrifugation, and to remove nucleic acids and intrinsically biotinylated proteins from the sample, 80 U ml$^{-1}$ benzonase nuclease (Santa Cruz Biotechnology) and 125 µg ml$^{-1}$ of avidin (Thermo Fisher Scientific) was added to the supernatant followed by a second round of centrifugation. The sample was then cleared by centrifugation, and the supernatants were poured into 10 ml gravity-flow columns (Bio–Rad) that were preloaded with 500 µl of Strep-Tactin beads (IBA GmbH) and allowed to drain under gravity. The beads were washed four times with 5 ml of lysis buffer without protease inhibitors (wash buffer). The purified proteins were eluted from the beads with 3 × 400 µl of low salt buffer (50 mM HEPES pH 8.0, 5 mM EDTA, 40 mM NaCl, 10 mM NaF) containing 0.3 mM biotin and concentrated at +4 °C to 25 µl volume using Amicon Ultra 10 kDa molecular weight cutoff centrifugal filters (Merck Millipore).

To stabilize the Commander complex by crosslinking, 2 mM bis(sulfosuccinimidyl)suberate (Thermo Scientific) was added for the final 30 min of the concentration. The crosslinked complex was purified in size-exclusion chromatography, using Superose 6 column in an ÄKTA Pure purification system using low salt buffer as eluent.

Samples were supplemented with n-decy-β-D-maltoside at a final concentration of 0.85 mM. Final protein concentration was 0.5 mg ml$^{-1}$

(native Commander) or 0.1 mg ml⁻¹ (crosslinked Commander). A 3 µl aliquot of sample was applied on a glow-discharged Quantifoil Cu 200 mesh R1.2/1.3 grid, followed by incubation period and plunge-freezing in liquid ethane using a VitroBot Mark IV cryoplunger (Thermo Fisher Scientific). A blot time of 7 s at 100% relative humidity and 6 °C temperature was used with a blot force of −15. Crosslinked Commander grids were prepared using Quantifoil Cu 200 mesh R1.2/1.3 grids with an additional 2 nm amorphous carbon support film and an incubation time of 5 min on the grid. The grids were stored in liquid nitrogen for subsequent screening and imaging.

## Cryo-EM data collection

Data for the native Commander complex was collected at the Umeå Core Facility for Electron Microscopy, Sweden, using a Titan Krios 300 kV transmission electron microscope (Thermo Fisher Scientific), with an X-FEG electron source and a Gatan K2 direct electron detector camera equipped with a BioQuantum energy filter. Data was collected at 1.5 to 3.0 µm defocus, 40 frames at 42.8 e⁻/Å² total electron exposure and a pixel size of 0.82 Å for a total of 5,884 movies collected. Energy filter slit width was set at 20 eV. For crosslinked Commander, two datasets were collected at the cryo-EM Swedish Infrastructure Unit of SciLifeLab, Stockholm, Sweden, in two separate sessions from two grids prepared identically in the same session, using a Titan Krios 300 kV transmission electron microscope (Thermo Fisher Scientific), with an X-FEG electron source and a Gatan K3 direct electron detector camera equipped with a BioQuantum energy filter set at a 20 eV slit width. Dataset 1 was collected at 0.4 to 2.2 µm defocus, 50 frames at 59 e⁻/Å² total electron exposure and a pixel size of 0.846 Å for a total of 20,675 movies collected. Dataset 2 was collected at 0.6 to 2.0 µm defocus, 45 frames at 56 e⁻/Å² total dose and a pixel size of 0.862 Å for a total of 35,094 movies collected. Data collection parameters are listed in Table 1.

## Cryo-EM data processing

Cryo-EM data were processed in cryoSPARC v.4.1.1 unless stated otherwise[84]. Native Commander complex was processed in parallel with the crosslinked dataset 1 (below and Extended Data Fig. 2c). Movies were motion corrected and contrast transfer function (CTF) parameters estimated using patch motion correction and Patch CTF Estimation in cryoSPARC, respectively. After manual curation of micrographs, 5,748 exposures were retained. Particles were initially picked using template picking, using an initial volume from preliminary analysis carried out with crosslinked dataset 1 as a picking template. A total of roughly 867,000 particles were picked and cleaned by four rounds of two-dimensional (2D) classification, resulting in roughly 127,000 particles retained. This particle set was binned by a factor of two and refined by one round of ab initio reconstruction using four classes, followed by heterogeneous refinement against the ab initio volumes and homogeneous refinement of the class with highest resolution features (~51,000 particles, 6.8 Å resolution, Nyquist limited). This set was further refined by a similar round of ab initio reconstruction using four classes followed by heterogeneous refinement and homogeneous refinement of the top class (~17,000 particles, 6.8 Å resolution, Nyquist limited). This was followed by per-particle motion correction[85] using unbinned particles and homogeneous refinement (~17.000 particles, 3.7 Å resolution). The resulting particles were used to train a TOPAZ model[86] and repick the micrographs. A total of around 340,000 particles were extracted, followed by a single round of 2D classification and removal of duplicate picks, which yielded a set of roughly 137,000 particles. Heterogeneous refinement against four copies of the final volume from the initial processing yielded a model with roughly 90,000 particles and 3.3 Å resolution from nonuniform refinement[87]. Particles were further unbinned and local motion correction was applied, which resulted in a consensus map at 3.2 Å resolution and an estimated B-factor of 66.

For crosslinked Commander, datasets 1 and 2 were processed separately until the final consensus refinement stage (Extended Data Fig. 2d). Before the final processing workflow, an initial exploratory analysis was carried out on dataset 1 where particles were picked using the supervised picking routine of xmipp3 from micrographs denoised using the Noise2Noise implementation in crYOLO[88] within the Scipion framework[89]. These particles were then processed in cryoSPARC by iterative cycles of multi-class ab initio and heterogeneous refinement to yield an initial model that encompassed the maximum volume of the complex at the cost of resolution (6.6 Å, ~24,000 particles). The final particles were used to train a TOPAZ neural network model for auto-picking particles in both datasets 1 and 2.

Movies were motion corrected and CTF parameters estimated using patch motion correction and patch CTF estimation implementations in cryoSPARC. Micrographs were manually curated to remove bad micrographs with a resulting micrograph count of 19,630 for dataset 1. The TOPAZ network trained using the initial processing was used to pick roughly 2.9 million particles. Particles were extracted in a box of 560 × 560 pixels at full size and downsampled to 140 × 140 pixels for 2D classification. Two rounds of 2D classification were carried out to classify the particles into 100 classes, with roughly 872,000 particles retained. The resulting particles were re-extracted in a box of 280 × 280 pixels and used in a six-class ab initio reconstruction followed by heterogeneous refinement using the six ab initio models as template volumes. One class containing the highest resolution features was selected, containing roughly 397,000 particles. These particles were used at full size for nonuniform refinement, optimizing per-particle scales, defoci and per-group CTF parameters with tilt and trefoil fits.

Dataset 2 was processed similarly to dataset 1 (Extended Data Fig. 2d). Pixel size set to 0.846 Å/pix to facilitate merging of the data (calibrated pixel size 0.861 Å/pix). After micrograph curation, 33,879 micrographs were retained. TOPAZ picking yielded roughly 4.7 million particles, reduced to roughly 4.5 million after one round of 2D classification. Heterogeneous refinement using the six ab initio volumes from dataset 1 yielded a single class with high-resolution features, with around 1.0 million particles retained. This class was first refined using nonuniform refinement and further polished by a round of four-class ab initio reconstruction followed by heterogeneous refinement. The final particle stack, containing roughly 577,000 particles, was merged with the particles from dataset 1 (total of ~958,000 particles) and refined using nonuniform refinement with per-particle defocus optimization and per-group CTF parameter optimization, fitting tilt and trefoil parameters and per-particle scales. The combined particles were polished by heterogeneous refinement against the current consensus map and six ab initio maps from the zeroth iteration against the data used as junk sinks. The particles that ended up in the consensus class (~667,000) were refined locally with a mask encompassing the COMMD-ring. The final gold-standard global Fourier-shell correlation (FSC) resolution was estimated at 2.9 Å with an estimated B-factor of 77. Local motion correction was performed but this showed no improvement in resolution.

To find the substructure of the bottom half of Commander (focused map 1, Extended Data Fig. 2f), partial signal subtraction was carried out using the final COMMD-ring map and a mask encompassing the COMMD-ring region of the map, using a low-pass filter of 10 Å with the final particle set. The resulting subtracted particles were refined locally using a Gaussian prior for poses and shifts at standard deviation of the prior over rotation of 30° and over shifts of 14 pix with FSC noise substitution enabled, a maximum alignment resolution of 12 Å imposed and using a batch size of 5,000, resulting in a map with nominal resolution of 8.6 Å. This particle set was then polished with several rounds of heterogeneous refinement against six junk sink classes, followed by local refinements, to reach a final particle stack of roughly 125,000 particles at a nominal resolution of 6.6 Å.

To resolve the Retriever substructure (focused map 2, Extended Data Fig. 2g), partial signal subtraction was carried out using the final

COMMD-ring map and a mask encompassing the I-coil, half of R-coil and DENND10 regions of the map using a low-pass filter at 10 Å with the final particle set. The resulting particles were refined locally using a Gaussian prior for poses and shifts at standard deviation of the prior over rotation of 20° and over shifts of 10 pix, with FSC noise substitution enabled and with a batch size of 5,000, resulting in a map with nominal resolution of 7.8 Å. The initial model was the coils map obtained from the previous step. This map was then used as an input in heterogeneous refinement together with six junk sinks as described for the consensus map to remove bad particles, resulting in a particle stack of roughly 195,000 particles. This particle set was refined locally using settings as above to reach a resolution of 7.6 Å. Then, 3D classification with ten classes, 250 Å high-pass filter, O-EM batch size of 2,000 per class and principal components analysis initialization mode with default settings was performed, and the class with most features at the VPS26C end of the map was selected. This particle set of around 13,000 particles was locally refined with an additional 10 Å maximum alignment resolution to prevent overfitting, reaching a nominal resolution of 7.5 Å.

The 3D variability analysis was run as implemented in cryoSPARC[90]. Particles used in reconstruction of focused map 1 were used without subtraction and with 3D alignments from the consensus map reconstruction. Then 20 volumes along each principal component were reconstructed using the 'intermediates' option.

## Model building

Atomic models were built into the crosslinked Commander cryo-EM density using AF2 models as a starting point in ChimeraX[91] and adjusted using ISOLDE[92]. AF2 models were generated using a local installation of ColabFold v.1.5.2 (ref. 93) running AlphaFold v.2.3.1 (ref. 94) and Alpha-Fold Multimer v.3 (ref. 95). Predictions were run using ten recycles, one ensemble, with dropouts activated, a max-seq of 128 and max-extra-seq 256 with relaxation disabled. AF2 output models were fitted into density as rigid bodies, and predictions or parts of predictions most closely matching the density were selected for model building. The top half of Commander was predicted using full sequences of COMMD proteins as well as residues 120–392 of CCDC22 and 21–377 of CCDC93. The CCDC sequences included the I-coil region, which was necessary for correct placement of α15 and α16 of CCDC22, although the orientation of the I-coil itself in these predictions showed no correspondence to the cryo-EM density. The initial models for DENND10 and the I-, R- and V-coils were obtained from two separate predictions (different random seed) from the same run where the sequences of CCDC22 (residues 1–110, 332–627), CCDC93 (residues 313–631), DENND10, VPS29, VPS35L and VPS26C were used as input. Output models were split such that the DENND10 side of the model (Extended Data Fig. 4a) contained residues 313–425 of CCDC93, residues 322–436 of CCDC22 and DENND10, while the V-coil side of the model (Extended Data Fig. 4b) contained residues 426–631 of CCDC93 and residues 1–110 and 437–627 of CCDC22. The structure of the Retriever subcomplex was predicted separately (Extended Data Fig. 4c). For parts in lower resolution areas of the map (NTDs of COMMD1, COMMD7, COMMD9 and COMMD10, CCDC22 and CCDC93 in the lower half, DENND10 and Retriever) were modeled in ISOLDE with corresponding AF2 input models as distance and torsion restraints, with predicted-alignment error weighting enabled. The ISOLDE-adjusted models were refined in real space in PHENIX[96] against sharpened maps, using the parameters created in ISOLDE with command isolde write phenixRsrInput. Structures were validated using Phenix and MolProbity[97]. Figures were made using ChimeraX[91]. For regions with worse than 4 Å resolution, poly-alanine chains were modeled and side chain information is presented in the figures for visualization purposes only. Model refinement parameters are in Table 1.

## Ragdoll representation of Commander complex models

Simplified representations of Commander complex regions for Fig. 5 were prepared using a combination of Bsoft[98] and ChimeraX[91].

Bsoft beditimg command was used to generate appropriately sized primitive shapes, which were then superposed on the respective structural models by generating simulated density maps with the ChimeraX command molmap for the appropriate regions. For COMMD-ring the superposition was based on the COMMD domains only. For the Retriever subcomplex, the VPS26C and N-terminal part of VPS35L (residues 114–406) was used to superpose one sphere, VPS29 for the other sphere and the C-terminal part of VPS35L (residues 408–739) for the cylinder. The I-coil cylinder was superposed on residues 310–377 of CCDC93, and 322–389 of CCDC22. The R-coil cylinder was superposed on residues 380–427 and 441–484 of CCDC93, and 396–481 of CCDC22. The V-coil trapezoidal prism was superposed on residues 491–606 of CCDC93, and 1–108 and 489–615 of CCDC22. The relative rotation angles were calculated using ChimeraX command align using Cα atoms of the underlying atomic coordinates.

## Reporting summary

Further information on research design is available in the Nature Portfolio Reporting Summary linked to this article.

## Data availability

For protein structures, the atomic coordinates and their corresponding cryo-EM maps have been deposited in the PDB with accession codes 8P0V, 8P0W and 8P0X, or in the EMDB with accession codes EMDB-17339, EMDB-17340 and EMDB-17341 for the COMMD-ring with CCDC22 and CCDC93, the I-, R-, V-coils and the Retriever subcomplex, and the V-coil and Retriever subcomplex, respectively. Additionally, the native Commander complex dataset has been deposited in the EMDB with accession code EMDB-17342. Proteomics data have been deposited in MassIVE database, under accession code MSV000091490. Source data used to generate the figures are provided with this paper.

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

## Acknowledgements

We thank S. Keskitalo, A. Tuhkala, P. Laurinmäki, B. Löflund, D. Morado and K. Walldén for technical assistance, and P. Lappalainen and V. Paavilainen for critical reading of the manuscript. The facilities and expertise of the HiLIFE Proteomics and cryo-EM units at the University of Helsinki, a member of Instruct-ERIC Centre Finland, FINStruct and Biocenter Finland are gratefully acknowledged. The cryo-EM data was collected at the cryo-EM Swedish National Facility funded by the Knut and Alice Wallenberg, Family Erling Persson and Kempe Foundations, SciLifeLab, Stockholm University and Umeå University. We acknowledge CSC – IT Center for Science, Finland, for generous computational resources. This study was supported by grant no. 314669 from the Research Council of Finland (to J.T.H.) and grant nos. 288475, 319303 and 336470 from the Research Council of Finland (to M.V.) The funders had no role in study design, data collection and analysis, decision to publish or preparation of the manuscript.

## Author contributions

The project was conceptualized by J.T.H. and M.V. S.L. carried out the cloning, cell work and protein purification. S.L. prepared MS samples. S.L. and E.-P.K. prepared the cryo-EM samples. E.-P.K. processed the cryo-EM data and built the complex model. J.T.H. supervised cryo-EM work. S.L. and M.V. analyzed the MS data. The original draft was written by E.-P.K. and M.V. All authors reviewed and edited the manuscript.

## Competing interests

The authors declare no competing interests.

## Additional information

**Extended data** is available for this paper at https://doi.org/10.1038/s41594-024-01246-1.

**Correspondence and requests for materials** should be addressed to Juha T. Huiskonen or Markku Varjosalo.

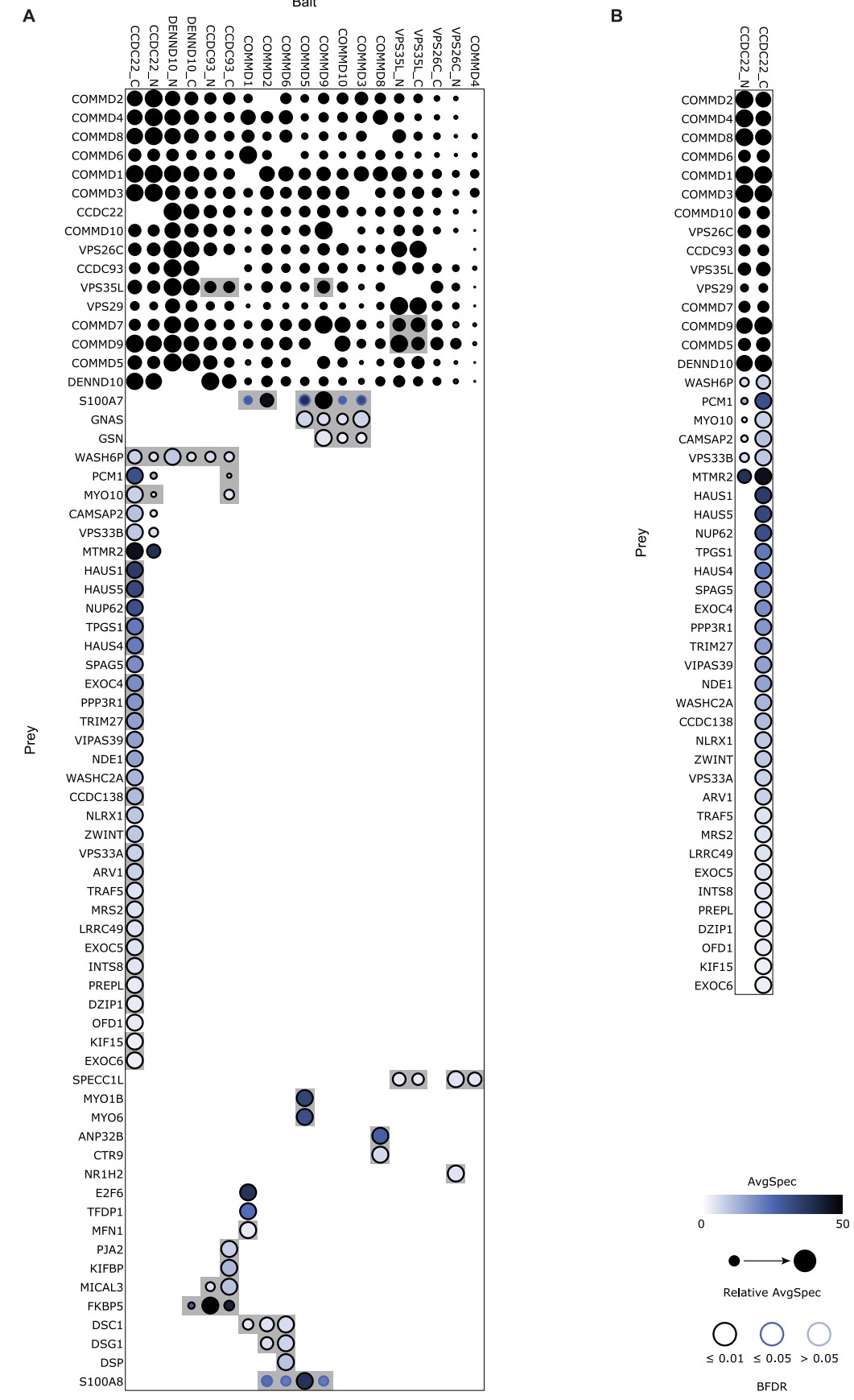

**Extended Data Fig. 1 | See next page for caption.**

**Extended Data Fig. 1 | Affinity purification mass spectrometry analysis of Commander components, related to Fig. 1. a,** Dot-plot visualization (BFDR ≤ 0.05) of the Commander complex proteins' interactors detected by the AP-MS. Each node corresponds to the abundance of the average spectral count for each prey, and the node size relative abundance of the prey. BFDR values are indicated with circles around the node, and previously uncharacterized interactions are highlighted with gray background. **b,** Focused Dot-plot visualization of the CCDC22 interactions. Commander complex components, WASH complex components, and proteins involved in WASH complex recruitment are indicated in bold typeface.

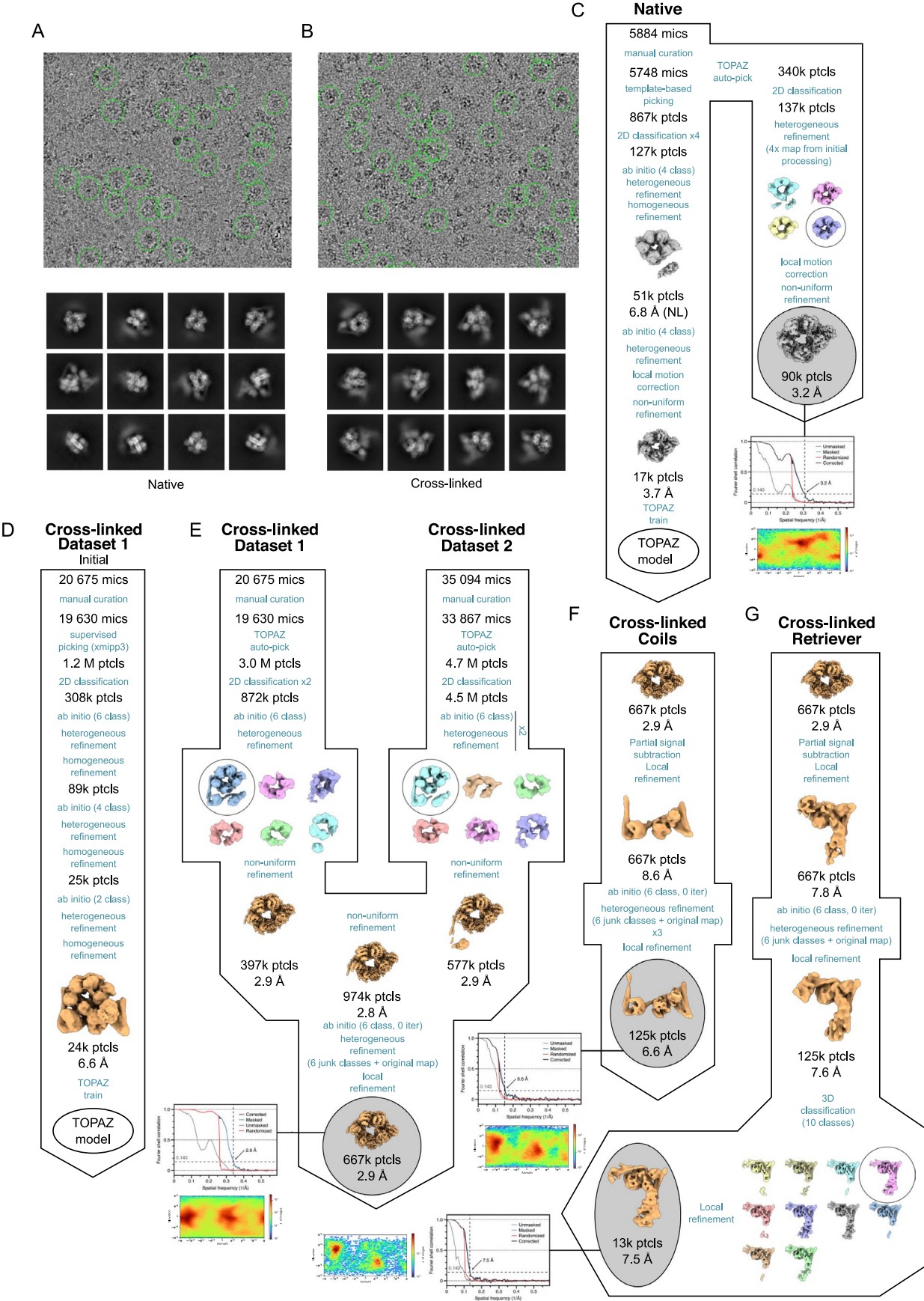

**Extended Data Fig. 2 | See next page for caption.**

**Extended Data Fig. 2 | Cryo-EM of native and crosslinked Commander, related to Fig. 2. a-b**, Representative cryo-EM micrographs and selected 2D class averages of **a** native and **b** crosslinked Commander. Particle images have been low-pass filtered to 4 Å and show particles picked for the consensus map reconstruction. **c-g**, Cryo-EM data processing workflow for **c**, native Commander, **d** preliminary processing of crosslinked Commander dataset 1, **e**, final processing of crosslinked Commander datasets 1 and 2 consensus maps, **f**, focused map 1 of crosslinked Commander datasets (coiled coil region), and **g**, focused map 2 of crosslinked Commander datasets (Retriever subcomplex).

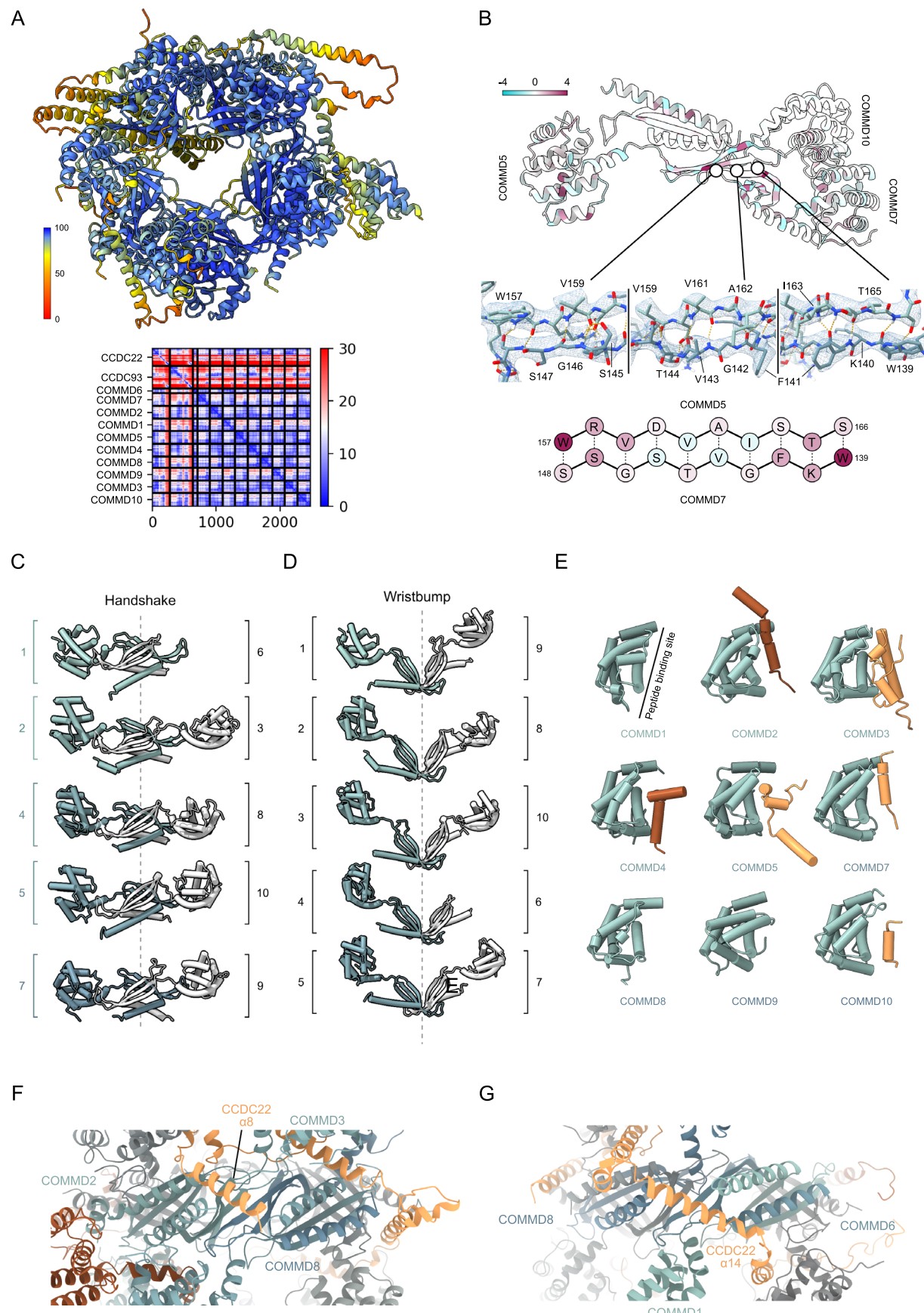

**Extended Data Fig. 3 | See next page for caption.**

**Extended Data Fig. 3 | Molecular models of Commander complex top half, related to Fig. 3. a**, AF2 prediction and the predicted alignment error (PAE) plot of the top half of the Commander complex, constituting the full sequences of COMMDs and residues 120–392 of CCDC22 and residues 21–377 of CCDC93. **b**, *Top*: Example wristbump interface between COMMD5 and COMMD7. *Middle*: three closeups of the model in cryo-EM density, highlighting the residues involved in the wristbump interaction interface between COMMD5 and COMMD7. *Bottom*: schematic representation of the example wristbump interface. Coloring is by sequence conservation within the human COMMD proteins in Top and Bottom subpanels. **c-d**, Structural models of all **c**, handshake and **d**, wristbump interactions. **e**, Models of NTDs of COMMD proteins (except COMMD6) depicted alongside parts of CCDC93 or CCDC22 that interact with them at the peptide binding site. **f-g**, Detail of **f** CCDC22 α8 or **g** CCDC22 α14 binding site on the COMMD-ring.

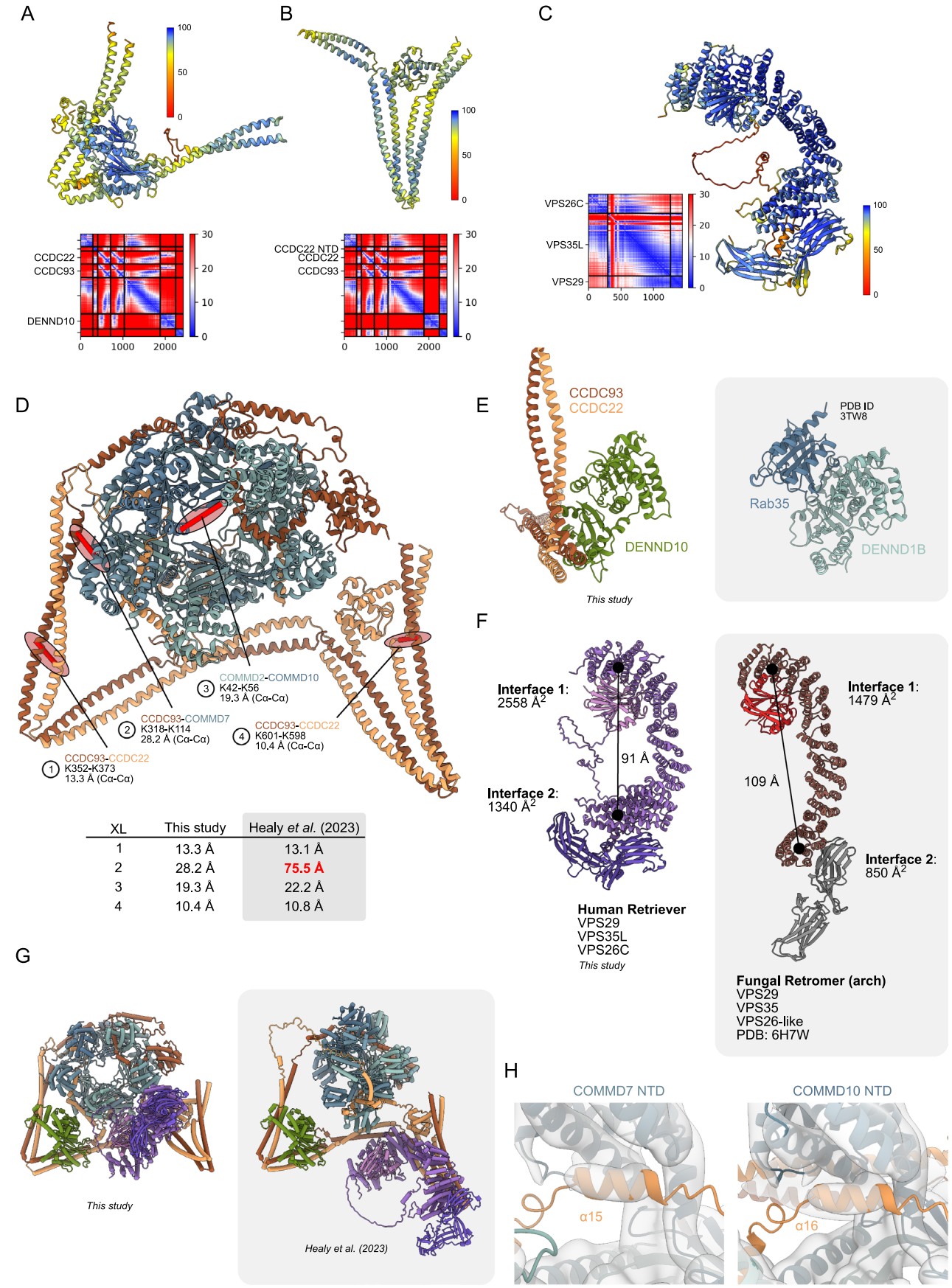

**Extended Data Fig. 4 | See next page for caption.**

**Extended Data Fig. 4 | Molecular models of Commander complex bottom half, related to Fig. 4. a**, AF2 model of DENND10 region of the Commander complex used as an initial model with predicted alignment error plots indicating the relevant chains. Model is colored according to per-residue pLDDt scores. Model has been trimmed based on the fit to the cryo-EM density map. AF2 model contained all chains of the bottom half during prediction. **b**, V-coil region of Commander as in **a**, with different random seed in AF2 prediction. **c**, Retriever subcomplex model as in **a. d**, Chemical crosslinks identified by MS in the context of the Commander structure model. **e**, Comparison of DENND1B-Rab35 complex structure (PDB 3TW8) with DENND10 in the context of Commander. I-coil sterically blocks the putative Rab binding site on DENND10. **f**, Structure of Retriever in the context of the Commander complex compared to the Fungal retromer structure. *Interface 1*: VPS29-VPS35(L). *Interface 2:* VPS35(L)-VPS26(C). Retriever adopts a contracted conformation compared to retromer and exhibits larger interaction interfaces. **g**, Comparison of Commander complex models presented in this study (left) and in Healy *et al. (right), kindly provided by the authors, superposed via DENND10.* **h**, *Density supporting the placement of CCDC22 α15 and α16. Map (EMD*-17340*) was low-pass filtered to 7 Å.*

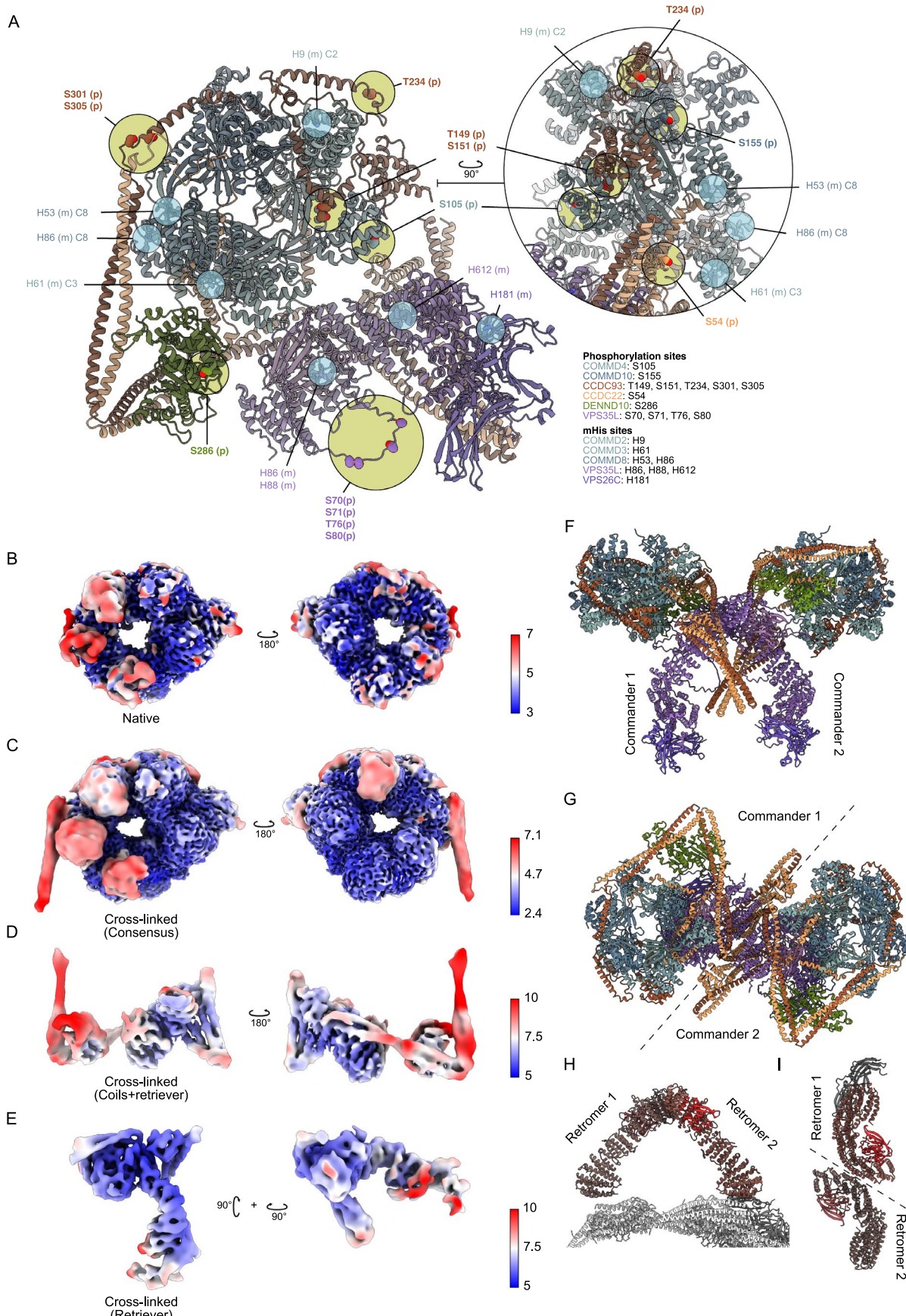

**Extended Data Fig. 5 | See next page for caption.**

**Extended Data Fig. 5 | Detected post-translational modifications, local resolution estimates and putative dimerization mode of Commander complex, related to Figs. 1, 3, and 4. a**, Molecular model of Commander complex with all detected phosphorylation and histidine methylation sites. m: met-His site, p: phosphorylation site. Inset: rotated model showing details on the CCDC93 NN-CH domain side of the complex. **b-e**, Local resolution estimates of cryo-EM maps from **b** native Commander, **c** crosslinked Commander consensus map **d** crosslinked Commander focused map 1 (Coils) and **e** crosslinked Commander focused map 2 (Retriever). Color bar indicates resolution in Å. **f**, Model of putative head-to-head dimerization of Commander complex prepared by superposition via VPS29 and VPS35(L) C-terminal region. **g**, Top view of the model in **f. h**, Retromer arch model (PDB ID 6H7W) depicted in the same orientation as Commander dimer model in **f. i**, Top view of the model in **h**. Models in **h-i** color-coded as in Extended Data Fig. 4f.

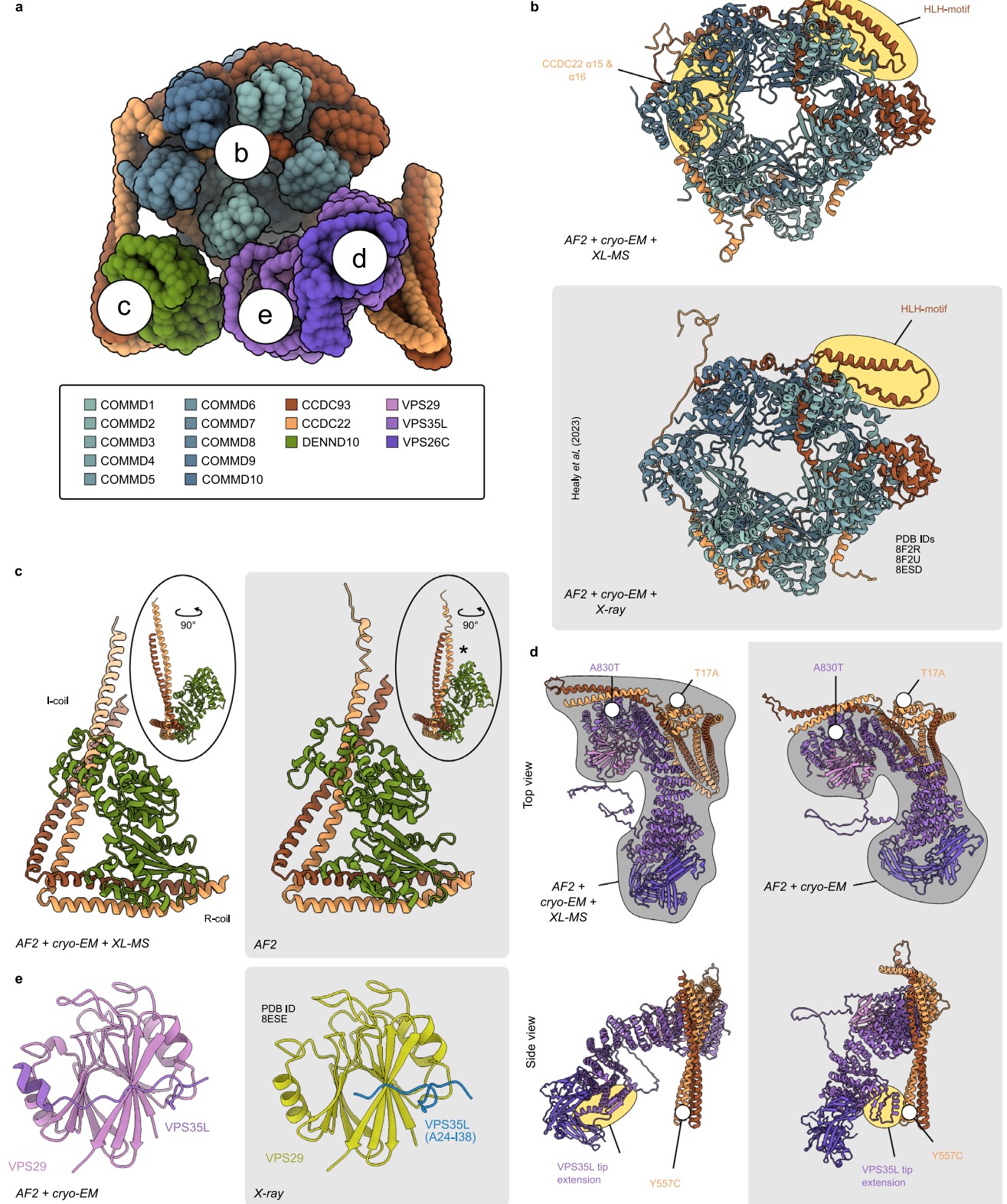

**Extended Data Fig. 6 | Comparative analysis of conformational variation in the Commander complex structure compared to existing literature, related to Fig. 5. a**, Overview of the Commander complex structure with location of following panels indicated. Comparison of **b** the COMMD-ring, **c** DENND10, I-coil, and R-coil region, **d** Retriever subcomplex from the structure presented in this study and the overall model presented by Healy et al. **e**, Comparison of VPS29 with VPS35L (13-37) presented in this study (left) and crystal structure of VPS29 with VPS35L (16-38) peptide (right). Major structural differences are highlighted with yellow, and sources of structural data are indicated for each structure. The three disease mutations analyzed in AP-MS and BioID (Fig. 7) are indicated in **d**.

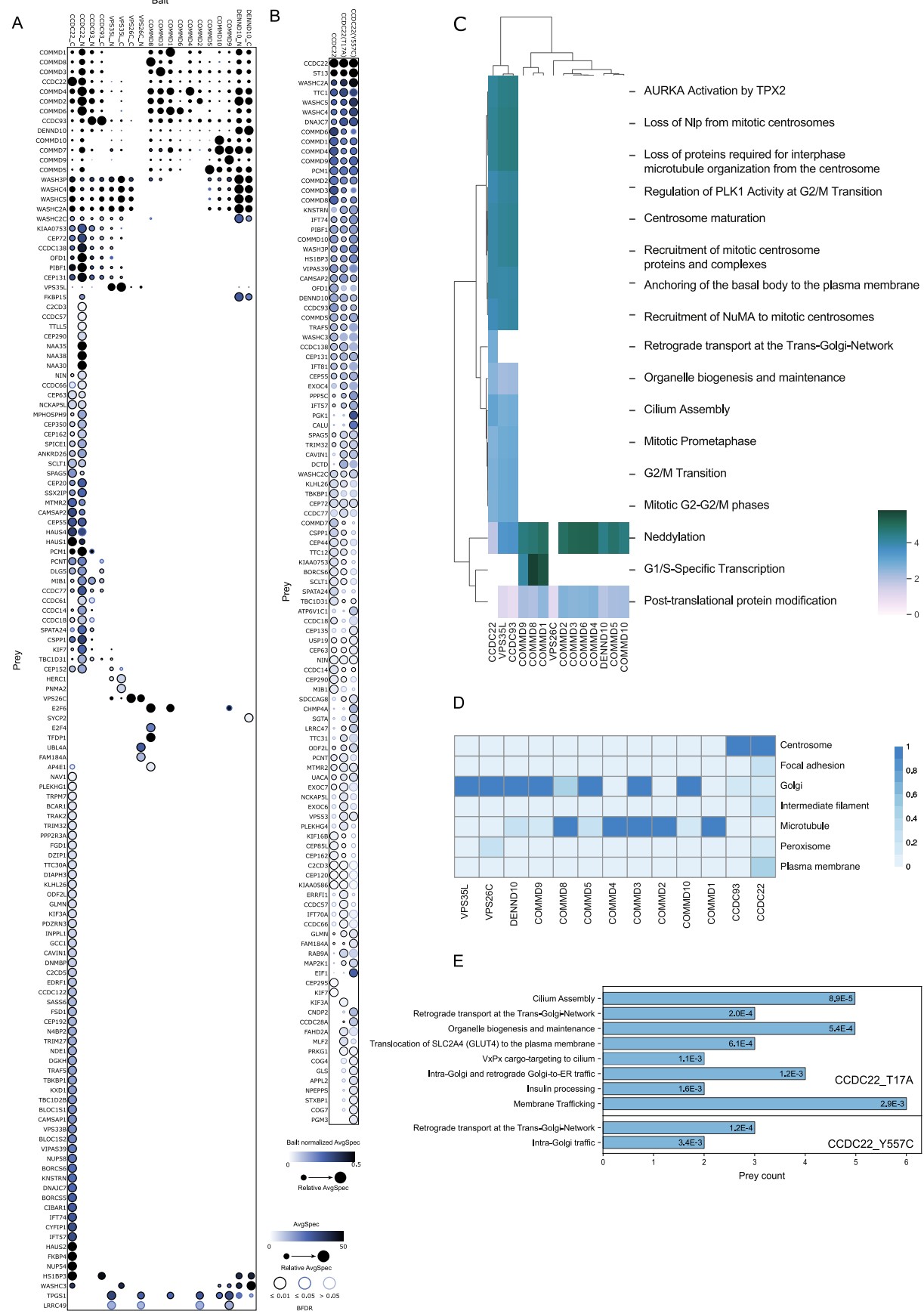

**Extended Data Fig. 7 | See next page for caption.**

**Extended Data Fig. 7 | Molecular interactors, context, and cellular pathways connected with individual Commander complex components, related to Fig. 6. a**, Dot-plot visualization (BFDR ≤ 0.05) of interactors of the Commander complex detected by the BioID-MS. Node color corresponds to the abundance of the average spectral count for each prey, and node radius to its relative abundance. **b**, Dot-plot visualization of RSS syndrome related point mutants of CCDC22 analyzed by BioID-MS. All PPIs passing HCI criteria to any of the CCDC22 variants are plotted, with HCIs are indicated with black outline and non-HCIs with light blue. Node color corresponds to the bait normalized abundance of the average spectral count for each prey, and node radius to its relative abundance across all baits determined by ProHits-Viz. **c**, Reactome pathways enriched for the Commander complex proteins. **d**, Molecular level localization of the Commander complex proteins obtained by MS-microscopy. **e**, Reactome pathways enriched (p < 0.005, values marked in bars) for the RSS disease variant HCIs distinct from the wild-type CCDC22.

# Reporting Summary

## Statistics

For all statistical analyses, confirm that the following items are present in the figure legend, table legend, main text, or Methods section.

| n/a | Confirmed | |
|---|---|---|
| ☐ | ☒ | The exact sample size (*n*) for each experimental group/condition, given as a discrete number and unit of measurement |
| ☐ | ☒ | A statement on whether measurements were taken from distinct samples or whether the same sample was measured repeatedly |
| ☒ | ☐ | The statistical test(s) used AND whether they are one- or two-sided<br>*Only common tests should be described solely by name; describe more complex techniques in the Methods section.* |
| ☒ | ☐ | A description of all covariates tested |
| ☒ | ☐ | A description of any assumptions or corrections, such as tests of normality and adjustment for multiple comparisons |
| ☐ | ☒ | A full description of the statistical parameters including central tendency (e.g. means) or other basic estimates (e.g. regression coefficient) AND variation (e.g. standard deviation) or associated estimates of uncertainty (e.g. confidence intervals) |
| ☒ | ☐ | For null hypothesis testing, the test statistic (e.g. *F*, *t*, *r*) with confidence intervals, effect sizes, degrees of freedom and *P* value noted<br>*Give P values as exact values whenever suitable.* |
| ☒ | ☐ | For Bayesian analysis, information on the choice of priors and Markov chain Monte Carlo settings |
| ☒ | ☐ | For hierarchical and complex designs, identification of the appropriate level for tests and full reporting of outcomes |
| ☒ | ☐ | Estimates of effect sizes (e.g. Cohen's *d*, Pearson's *r*), indicating how they were calculated |

*Our web collection on statistics for biologists contains articles on many of the points above.*

## Software and code

Policy information about availability of computer code

| | |
|---|---|
| Data collection | CryoEM data was collected using EPU software (version 2.11.0, Thermo-Fisher). The MS data was collected in the study using commercial Bruker Daltonics timsControl (version 4.1.12) & HyStar (version 6.2.1.13) software. |
| Data analysis | CryoEM data was analysed principally using cryoSPARC (v. 4.1.1), with additional particle picking routines performed using TOPAZ (version 0.2.4). Initial data processing was carried out using implementations of RELION (v. 3.1) and crYOLO (v. 1.8.0) within the Scipion framework and utilities therein.<br>Maps and models were analyzed using ChimeraX (v. 1.5), and models built using a combination of ISOLDE (v. 1.5) and PHENIX (version 1.21). Initial models were obtained by prediction using AlphaFold v. 2.3.1 and AlphaFold-Multimer (v3) via local installation of ColabFold (v. 1.5.2).<br><br>CRAPome, Mellacheruvu et al., 2013 http://www.crapome.org/<br>Cytoscape version 3.1.0, Shannon et al., 2003 http://www.cytoscape.org/<br>DAVID Bioinformatics Resources, National Institute of Allergy and Infectious Diseases (NIAID), NIH https://david.ncifcrf.gov/home.jsp<br>Gene Ontology analysis tool DAVID, Bioinformatics Resources 6.8 National Institute of Allergy and Infectious Diseases (NIAID), NIH https://david.ncifcrf.gov/home.jsp<br>Hierarchial clustering tool Biohit-viz and Pro-Hits-viz, Gigras lab, Department of Molecular Genetics, University of Toronto, Toronto, Ontario, Canada https://prohits-viz.lunenfeld.ca<br>Interaction network analysis tool: PINA v2.0, Cowley et al., 2012 http://cbg.garvan.unsw.edu.au/pina/<br>SAINTexpress version 3.1.0, Choi et al., 2011, http://saint-apms.sourceforge.net/Main.html |

For manuscripts utilizing custom algorithms or software that are central to the research but not yet described in published literature, software must be made available to editors and reviewers. We strongly encourage code deposition in a community repository (e.g. GitHub). See the Nature Portfolio guidelines for submitting code & software for further information.

## Data

Policy information about <u>availability of data</u>

All manuscripts must include a <u>data availability statement</u>. This statement should provide the following information, where applicable:
- Accession codes, unique identifiers, or web links for publicly available datasets
- A description of any restrictions on data availability
- For clinical datasets or third party data, please ensure that the statement adheres to our <u>policy</u>

Databases used in the study:
Mammalian protein complex resource: CORUM Institute of Bioinformatics and Systems Biology, Helmholtz Zentrum München, http://mips.helmholtz-muenchen.de/corum/
Uniprot: https://www.uniprot.org/

Electron microscopy maps are deposited in the electron microscopy data bank (EMDB) with accession codes EMD-17340, EMD-17339, EMD-17341 and EMD-17342. Protein structure models were deposited into the protein data bank (PDB) with accession codes 8POW, 8POV and 8POX.

The source data of the figures are provided with this paper as a separate Excel sheet. The MS peptide raw data from the MS runs have been deposited in the Massive database (https://massive.ucsd.edu/ProteoSAFe/private-dataset.jsp?task=985dbde9f67146758bbd959b3847703f) under accession number MSV000091490

Material Availability
Plasmids generated in this study will be deposited in Addgene. No other unique reagents were generated in this study.

## Research involving human participants, their data, or biological material

Policy information about studies with <u>human participants or human data</u>. See also policy information about <u>sex, gender (identity/presentation), and sexual orientation</u> and <u>race, ethnicity and racism</u>.

| | |
|---|---|
| Reporting on sex and gender | N/A |
| Reporting on race, ethnicity, or other socially relevant groupings | N/A |
| Population characteristics | N/A |
| Recruitment | N/A |
| Ethics oversight | N/A |

Note that full information on the approval of the study protocol must also be provided in the manuscript.

# Field-specific reporting

Please select the one below that is the best fit for your research. If you are not sure, read the appropriate sections before making your selection.

☒ Life sciences  ☐ Behavioural & social sciences  ☐ Ecological, evolutionary & environmental sciences

For a reference copy of the document with all sections, see nature.com/documents/nr-reporting-summary-flat.pdf

# Life sciences study design

All studies must disclose on these points even when the disclosure is negative.

| | |
|---|---|
| Sample size | Sufficient sample size (number of cryoEM images and extracted particles) was determined by collecting enough data that would result a reconstructed map at sufficient resolution (estimated by Fourier shell correlation). For other experiments sufficient sample sizes were chosen for each experiment to determine whether the outcome was statistically significant. For cell line experiments minimal number replicates for each experiments was two or more |
| Data exclusions | Micrographs that failed in contrast transfer function estimation were discarded. Particles that did not show features of the complex were discarded in 2D-classification step. |
| Replication | The data were divided in two random subsets and processed separately as part of CryoSPARC's gold standard refinement protocol. All attempts at replication were successful. When applicable, experiments were independently repeated minimally twice. |
| Randomization | CryoEM data were randomly split in two half data sets in the beginning of the image processing (gold standard refinement). Other type of |

| | |
|---|---|
| Randomization | randomisation is not relevant for cryoEM as data collected represents unbiased views of the sample. Different samples (different baits or different analysis method) were allocated to MS analysis in a random order |
| Blinding | Investigators were not blinded for any of the analyses as the analytical methods employed were automated and do not allow for subjective interpretation. |

# Reporting for specific materials, systems and methods

We require information from authors about some types of materials, experimental systems and methods used in many studies. Here, indicate whether each material, system or method listed is relevant to your study. If you are not sure if a list item applies to your research, read the appropriate section before selecting a response.

## Materials & experimental systems

| n/a | Involved in the study |
|---|---|
| ☒ | Antibodies |
| ☐ | ☒ Eukaryotic cell lines |
| ☒ | Palaeontology and archaeology |
| ☒ | Animals and other organisms |
| ☒ | Clinical data |
| ☒ | Dual use research of concern |
| ☒ | Plants |

## Methods

| n/a | Involved in the study |
|---|---|
| ☒ | ChIP-seq |
| ☒ | Flow cytometry |
| ☒ | MRI-based neuroimaging |

## Eukaryotic cell lines

Policy information about cell lines and Sex and Gender in Research

| | |
|---|---|
| Cell line source(s) | HEK293 Flp-In T-REx (Thermo-Fisher) |
| Authentication | The cell line was obtained directly from commercial sources or biological repositories; additionally only low passage cells (passage number <10) were used for experiments. |
| Mycoplasma contamination | cell lines tested negative for mycoplasma |
| Commonly misidentified lines (See ICLAC register) | no commonly misidentified cell lines used |

