## [Peer Review File · Nature Structural & Molecular Biology]

Peer Review Information

Manuscript Title: Structure and Interactions of the Endogenous Human Commander Complex

Corresponding author name(s): Markku Varjosalo, Juha Huiskonen

Reviewer Comments & Decisions:

Decision Letter, initial version:

Message: 11th Aug 2023

Dear Dr. Varjosalo,

Thank you again for submitting your manuscript "Structure and Interactions of the Endogenous Human Commander Complex". We now have comments (below) from the 2 reviewers who evaluated your paper. In light of those reports, we remain interested in your study and would like to see your response to the comments of the referees, in the form of a revised manuscript.

You will see that while reviewers appreciate the results, they raise several concerns which will need to be addressed in a revision. Specifically, we would ask you to restructure the manuscript, including comparative analysis, to highlight the novelty and significance of the presented data, as suggested by the reviewers. You will notice that while reviewer #1 noted the value of the structure of the native complex, as clearly different from the model presented by Healy et al, reviewer #2 points out lack of novelty in the structure itself. We ourselves had to look at the cryo-EM data and models to understand the advance, which is why we would recommend this to be rectified in the manuscript itself, clearly stating the difference of the native complex obtained in this study, and how it stands compared to published literature. While we agree with reviewer #1 that further functional analysis would strengthen the manuscript, in the interest of timeliness we do not consider it essential in the context of the current work, especially considering the added value of the interactome.

Please be sure to address/respond to all concerns of the referees in full in a point-by-point response and highlight all changes in the revised manuscript text file. If you have comments that are intended for editors only, please include those in a separate cover letter.

We are committed to providing a fair and constructive peer-review process. Do not hesitate to contact us if there are specific requests from the reviewers that you believe are

technically impossible or unlikely to yield a meaningful outcome. We are always happy to discuss revision plans over the phone if the requests are not clear. Please reach out to me if you would like to schedule a call.

We expect to see your revised manuscript within 6 weeks. If you cannot send it within this time, please contact us to discuss an extension; we would still consider your revision, provided that no similar work has been accepted for publication at NSMB or published elsewhere.

Reporting Summary:

When submitting the revised version of your manuscript, please pay close attention to our [href="https://www.nature.com/nature-portfolio/editorial-policies/image-integrity">Digital Image Integrity Guidelines. and to the following points below:](https://www.nature.com/nature-portfolio/editorial-policies/image-integrity)

Please note that all key data shown in the main figures as cropped gels or blots MUST be presented in uncropped form, with molecular weight markers. These data can be aggregated into a single supplementary figure item. While these data can be displayed in a relatively informal style, they must refer back to the relevant figures.

SOURCE DATA: we request that the authors provide, in tabular form, the data underlying the graphical representations used in figures. This is to further increase transparency in data reporting, as detailed in this editorial (<http://www.nature.com/nsmb/journal/v22/n10/full/nsmb.3110.html>). Spreadsheets can

be submitted in excel format. Only one (1) file per figure is permitted; thus, for multi-paneled figures, the source data for each panel should be clearly labeled in the Excel file; alternately the data can be provided as multiple, clearly labeled sheets in an Excel file. When submitting files, the title field should indicate which figure the source data pertains to.

Data availability: this journal strongly supports public availability of data. All data used in accepted papers should be available via a public data repository, or alternatively, as Supplementary Information. If data can only be shared on request, please explain why in your Data Availability Statement, and also in the correspondence with your editor. Please note that for some data types, deposition in a public repository is mandatory - more information on our data deposition policies and available repositories can be found below: <https://www.nature.com/nature-research/editorial-policies/reporting-standards#availability-of-data>

[Redacted]

Sincerely,

Katarzyna Ciazynska
(she/her)
Associate Editor
Nature Structural & Molecular Biology
<https://orcid.org/0000-0002-9899-2428>

Referee expertise:

Referee #1: cryo-EM, endosomal trafficking

Referee #2: cryo-EM, endosomal trafficking

Reviewers' Comments:

Reviewer #1:

Remarks to the Author:

Recently discovered Commander complex exerts its function in multiple processes; however, structural information is lacking to deconvolute the underlying mechanisms. The proposed manuscript elegantly combines mass-spectrometric analysis with cryo-electron microscopy to visualize the whole commander complex and systematically analyze its interactome. This approach is granted for a 16-subunit commander complex and, in the broader context, sets up a promising way to parallelize molecular and structural discoveries of challenging biological assemblies.

The authors first created cell lines with one of the commander subunits with MAC tag (BIR^A* biotinylation enzyme plus affinity purification peptides). They systematically explored subunits interactomes, cross-referencing Bio-ID and affinity purification mass spectrometry (AP-MS) to get an insight into the stability and composition of commander assembled from endogenously expressed properties (except the bait under control of inducible promotor). The verified reported and discovered a plethora of new binders. The binders list highlights multiple roles of the commander. It includes proteins working in the Cilium Assembly pathway, membrane trafficking (exocyst, PI3P phosphatase), inflammation, and cytoskeleton organization, to name a few functional clusters. Finally, the MS analysis pinpointed posttranslational modification sites.

Following the MS, the cell line with COMMD9 bait was used to isolate the natively assembled commander for cryo-EM structure determination. SPA analysis converged on the map resolving primarily the COMMD ring. Cross-linking was used to restrict the conformational heterogeneity of the complex and succeeded in resolving the whole commander complex. As expected, the COMMD ring map still led in resolution and resolved side chains; nevertheless, the CCDCs and Retriever modules of the commander could now be resolved to the secondary structure using focused alignment. Commander architecture from cryo-EM map drastically differs from the predicted models (Healy et al. 2023; Boesch et al. 2023), featuring ~90 degrees of the COMMD ring rotation and other major differences. This finding is also valuable as a benchmark for predicting complex

tertiary structures. Furthermore, the structural studies were complemented by molecular dynamic simulations pinpointing conformation variability in COMMD that may bridge retriever and DENND10.

Finally, the manuscript looks closely at the loosely associated interactors discovered by proximity biotinylation (BioID-MS). BioID-MS proteome is dominated by inter-complex binding reiterating on commander multiple roles. Distance dependence of biotinylation by BirA* allowed the identification of two interaction clusters in the commander surface.

Overall the manuscript brings important novel insights and is written compellingly with sufficient experiential evidence. Authors deserve special credit for clear visual language and concise text.

Here I propose text and figure changes to improve manuscript clarity further.

Figure1

Fig1A is a good entry illustration but can be made more explicit. I suggest adding a legend for the elements of MAC tag and affinity resins that allow understanding of which tag element is used, similar to the figures in your Liu et al 2018 paper.

The arrows from AP-MS and XL-MS branches point to a mass spectrometer (?), but the BioID-MS - to the PPI analysis icon. I suggest leaving only PPI analysis in and converging on it all three paths.

Outline conditions for harsh and soft lysis conditions in the figure, i.e. "harsh, 0.1% SDS", "soft, 0.5% IGEPAL".

Replace the SEC panel in panel A with panel E. Use text instead of pictograms of molecules for SEC peaks designation.

On the right from the Cryo-EM block, you could name and show all four maps from grey circles in FigS2.

Also, please include Coomassie stained PAGE of native and cross-linked Commander used for cryo-EM.

Fig 1D. Define table axes. If feasible, introduce a better graphical explanation of the colour/circle size code. Panel A can be used to reference specific branches of the MS.

Consider moving panel B to supplementary if more space is needed.

AvgSpec/Relative AvgSpec abundance needs a better explanation for readers outside the MS field. Consider extending the Quantification and statistical analysis chapter and explaining the interpretation of two marginal data points (i.e. for good and bad binders).

Line 142. Comment why COMMD9 was chosen as bait.

Lien 480. Cloning of Commander complex components. Describe cell genotype in plain language. If I understand correctly, cell lines contained an additional copy (over the WT genome) of genome-integrated MAC-tagged tagged bait under a tetracycline-inducible promoter. Upon induction, the binders will preferentially bind to the tagged due to its higher titer than WT copies. What is the estimated bait expression level compared to the endogenously expressed protein?

Line 91. Highlight these newly discovered interactors in Fig S1.

Line 567. Give protein concentration in the BS3 cross-linking reaction

FigS2. Add title "cross-linked" above panels D-G. Show FSC curves for the final maps.

Line 596. Should “tandem” be replaced with “in parallel”? Tandem implies sequential order.

Fig3J. Indicate the NN-CH domain in the figure. Should CCDC93 NTD be changed to CCDC93 NN-CH?

FigS4. Fix panel assignment (F panel is skipped). The comparison to Healy et al 2023 AF2 model is interesting and will be clearer using pipes and plunks depiction or “ragdoll model” with complex parts presented as geometrical shapes to highlight dramatic differences in COMMDs ring rotation and CCDCs coils. Aligning the models on Vps35L may be more informative.

Line 134. DENND10 binds the commander via CCDCs, making the statement in line 134 counterintuitive. Please clarify this point.

Boesch DJ et al. (2023) Structural Organization of the Retriever-CCC Endosomal Recycling Complex. bioRxiv. doi:10.1101/2023.06.06.543888
 Healy MD et al. (2023) Structure of the endosomal Commander complex linked to Ritscher-Schinzel syndrome. Cell 186:2219-2237 e2229. doi:10.1016/j.cell.2023.04.003

Reviewer #2:

Remarks to the Author:

Laulumaa et al. here report the use of AP-MS and BioID-MS to make a comprehensive characterization of stable and transient interactions of the human Commander complex. The authors propose a set of biological processes where Commander could have essential roles such as endosomal transport, actin nucleation, immune response, transcription regulation, centriole replication, centrosomal targeting and cilium assembly. In addition, the authors purified and solved the structure of the whole Commander complex by single particle cryo-EM. The structure consist of a 16-protein assembly arranged in two sub-complexes; an heterodecameric ring of the COMMD proteins (COMMD 1-10) connected to the retriever sub-complex (VPS35L, VPS26C and VPS29) by the CCDC22-CCDC93 heterodimer which interacts with DENND10 through a V-shaped coil (R-coil). The atomic model built into the cryo-EM density is supported by several XL-MS crosslinks. Overall, this is a well-executed study that provides the molecular architecture of the full Commander complex and spans its interactome. Unfortunately, the recent publication of the structure of the Commander complex (Healy et al. 2023) diminishes the originality of the present work. Nevertheless, while the structure of the Commander complex from Healy et al. was determined by combining data form cryo-EM, X-ray crystallography and AF modelling of distinct sub-structures, the present work by Laulumaa et al. provides a complete picture of the full sixteen subunits. Sadly, the authors have not done a clear comparative analysis between both structures putting an emphasis on aspects that were not presented/discussed by Healy et al. On the other hand, the interactome analysis reads as a dry catalog of interactions with assigned GO terms and biological processes. The manuscript lacks a logical flow where functional/biological mechanisms are addressed throughout the structure. Similarly, the lack of functional assays 'in cellulo' diminishes general interest. For example, assessing the integrity of the Commander complex throughout site directed mutagenesis and evaluating the significance on novel roles such

as cilia modulation could add novelty and interest to a wider audience. In summary, the lack of novelty in the structure and the absence of functional implications connected to Commander assembly have dumped the excitement for this manuscript.

Other more specific issues include:

- This reviewer has not been able to see the PDB validation report to assess the PDB model.
- Supplementary Table (REAGENT or RESOURCE) lacks several EMD and PDB codes. They are denoted as 'EMD-XXX' or 'PDB: XXXX'. Same in 'Data availability' (lines 710-718)
- There is a very early fall off in the FCS(unmasked). Could it be because of the very large box size used in relation to the size of the complex?. Was it intended to capture flexible regions?.
- In section 4.2 of the validation report, there is a big difference in the reported resolution and the unmasked-calculated resolution. Could this be related to the large box size used?
- It might be something odd from the validation report but the mask looks larger in 2.6.1 than the projection in 2.1 (although the map in 2.5.1 looks large like the mask, it looks different to the 2.5.2).

Author Rebuttal to Initial comments

Dear reviewers,

We thank you for your detailed and positive comments on our manuscript "Structure and Interactions of the Endogenous Human Commander Complex". See below our point-by-point response to your comments (in gray italics). Further, we have listed additional modifications and reported additional experiments at the end of this response.

Reviewer 1:

Remarks to the Author:

Recently discovered Commander complex exerts its function in multiple processes; however, structural information is lacking to deconvolute the underlying mechanisms. The proposed manuscript elegantly combines mass-spectrometric analysis with cryo-electron microscopy to visualize the whole commander complex and systematically analyze its interactome. This approach is granted for a 16-subunit commander complex and, in the broader context, sets up a promising way to parallelize molecular and structural discoveries of challenging biological assemblies.

The authors first created cell lines with one of the commander subunits with MAC tag (BirA biotinylation enzyme plus affinity purification peptides). They systematically explored subunits interactomes, cross-referencing Bio-ID and affinity purification mass spectroscopy (AP-MS) to get an insight into the stability and composition of commander assembled from endogenously expressed properties (except the bait under control of inducible promotor). The verified reported and discovered a plethora of new binders. The binders list highlights multiple roles of the commander. It includes proteins working in the Cilium Assembly pathway, membrane trafficking (exocyst, PI3P phosphatase), inflammation, and cytoskeleton organization, to name a few functional clusters. Finally, the MS analysis pinpointed posttranslational modification sites.*

Following the MS, the cell line with COMMD9 bait was used to isolate the natively assembled commander for cryo-EM structure determination. SPA analysis converged on the map resolving primarily the COMMD ring. Cross-linking was used to restrict the conformational heterogeneity of the complex and succeeded in resolving the whole commander complex. As expected, the COMMD ring map still led in resolution and resolved side chains; nevertheless, the CCDCs and Retriever modules of the commander could now be resolved to the secondary structure using focused alignment. Commander architecture from cryo-EM map drastically differs from the predicted models (Healy et al. 2023; Boesch et al. 2023), featuring ~90 degrees of the COMMD ring rotation and other major differences. This finding is also valuable as a benchmark for predicting complex tertiary structures. Furthermore, the structural studies were complemented by molecular dynamic simulations pinpointing conformation variability in COMMD that may bridge retriever and DENND10. Finally, the manuscript looks closely at the loosely associated interactors discovered by proximity biotinylation (BioID-MS). BioID-MS proteome is dominated by inter-complex binding reiterating on commander multiple roles. Distance dependence of biotinylation by

BirA allowed the identification of two interaction clusters in the commander surface.*

Overall the manuscript brings important novel insights and is written compellingly with sufficient experiential evidence. Authors deserve special credit for clear visual language and concise text.

Here I propose text and figure changes to improve manuscript clarity further.

Figure 1

Fig 1A is a good entry illustration but can be made more explicit. I suggest adding a legend for the elements of MAC tag and affinity resins that allow understanding of which tag element is used, similar to the figures in your Liu et al 2018 paper. The arrows from AP-MS and XL-MS branches point to a mass spectrometer (?), but the BioID-MS - to the PPI analysis icon. I suggest leaving only PPI analysis in and converging on it all three paths. Outline conditions for harsh and soft lysis conditions in the figure, i.e. "harsh, 0.1% SDS", "soft, 0.5% IGEPAL".

Thank you for your feedback on Figure 1.

In response to your suggestions, we have revised Fig 1A to enhance its clarity and understandability. We added labels for both the SH-tag and the biotin ligase BirA, aiming to provide clearer visualization of the involved elements.

We would like to clarify the representation of the Mass spectrometer and PPI analysis in Fig 1A. They are intended to represent the same analytical endpoint. To address the inconsistency you pointed out, we've adjusted the figure such that the arrow from BioID-MS now correctly points to the combined MS-analysis and PPI analysis icon. This should now provide a consistent representation of the analytical endpoints for the different techniques.

Regarding the 'lysis conditions', we appreciate your suggestion on detailing the lysis conditions directly in the figure. However, the distinction between soft and harsh lysis is nuanced, involving more than just the choice of detergent. Specifically, the harsh lysis procedure also incorporates sonication in the presence of benzonase. We believe that these intricacies are best detailed in the experimental section, where they can be elaborated upon more extensively. Hence, we've opted to maintain a streamlined representation in Fig 1A, without the specific detergent details, to ensure its simplicity and accessibility.

Replace the SEC panel in panel A with panel E. Use text instead of pictograms of molecules for SEC peaks designation.

To clarify, the SEC depiction in panel A is intended as a schematic representation, designed to provide an overview of the workflow employed in our study. It is not a direct representation of experimental results.

On the right from the Cryo-EM block, you could name and show all four maps from grey circles in FigS2.

Thank you for your feedback regarding the representation of the Cryo-EM block in our figure.

We understand the value in showcasing detailed results, as you've suggested with the four maps from grey circles in FigS2. However, the primary intent behind Figure 1 panel A was to provide a schematic overview of the entire workflow. Introducing specific results into this schematic might divert from its original purpose of giving a broad-strokes view of the methodology.

Also, please include Coomassie stained PAGE of native and cross-linked Commander used for cryo-EM.

Thank you for your suggestion to include a Coomassie stained PAGE of the native and cross-linked Commander used for cryo-EM.

While in general we appreciate the value of such data, here we've decided against this inclusion for the following critical reasons:

1. The procedure for purifying the cross-linked Commander complex results in a very limited yield (~25 μ l at ~0.1 mg/ml). We, therefore, prioritized cryo-EM grid preparation over PAGE analysis. We were confident in this approach, as we had extensive AP-MS evidence of the constituents of the complex before cross-linking and identified all components of the complex in the MS data we collected.
2. Furthermore, given the mild nature of our cross-linking procedure, we anticipate a set of heterogeneously cross-linked complex isoforms in a PAGE analysis. Such an analysis would primarily indicate that cross-linking has occurred, without providing intricate details. We can obtain more precise information about the cross-linking pattern from the MS data we collected from the cross-linked, gel-filtered complex.

Fig 1D. Define table axes.

Table axes (bait / prey) have now been added to Figure 1 panel D.

If feasible, introduce a better graphical explanation of the colour/circle size code.

Thank you for the feedback on our graphical representation.

To enhance clarity, we've renamed the colour/circle size codes to "Bait normalized AvgSpec" and "Relative AvgSpec." We've expanded on this in the figure legend with the

following explanation on Page 25, line 8: "The color of each circle represents the abundance of each prey normalized to the mean abundance of the bait protein, and the circle radius indicates the relative abundance across all samples calculated by ProHits-Viz."

We hope this adjustment provides a clearer understanding of the graphical elements used.

Panel A can be used to reference specific branches of the MS. Consider moving panel B to supplementary if more space is needed.

We have incorporated a reference to panel A in the text.

AvgSpec/Relative AvgSpec abundance needs a better explanation for readers outside the MS field. Consider extending the Quantification and statistical analysis chapter and explaining the interpretation of two marginal data points (i.e. for good and bad binders).

The average spectral count corresponds to the abundance of each prey protein in the sample and is represented by the node color. To evaluate the relative weight of each interaction among the presented samples, we calculated the relative abundance using the ProHits-Viz tool.

We have updated the Figure 1 legend on Page 25, line 8 to read: "The color of each circle represents the abundance of each prey, normalized to the mean abundance of the bait protein. The circle radius indicates the relative abundance across all samples, as calculated by ProHits-Viz."

Line 142. Comment why COMMD9 was chosen as bait.

As there was initially very little information about the Commander complex assembly, we selected a few bait proteins for purification optimization. Out of the tested bait proteins, COMMD9 gave the highest yield, and was thus chosen for further optimization.

Line 480. Cloning of Commander complex components. Describe cell genotype in plain language. If I understand correctly, cell lines contained an additional copy (over the WT genome) of genome-integrated MAC-tagged bait under a tetracycline-inducible promoter. Upon induction, the binders will preferentially bind to the tagged due to its higher titer than WT copies. What is the estimated bait expression level compared to the endogenously expressed protein?

The cell line used employs the Flp-In™ system, which allows the insertion of a single copy of the gene of interest at the FRT site in HEK cells. Expression of the protein of interest is

induced using tetracycline, resulting in an expression level parallel to and on the same scale as the endogenous protein [PMID:19156129; PMID:23455922; PMID:28330616]. In BioID-MS experiments, proximity labeling biotinylation is induced using biotin. Intact protein complexes (AP-MS) or biotinylated proteins (after disruption of native protein complexes in BioID-MS) are extracted from the cell lysate using Strep-tactin resin, which binds both biotin and the SH-tag. The method is described in more detail in the articles by Liu et al [PMID:29568061], which we reference in the text on Page 12, line 30, and on Page 13, line 6.

Line 91. Highlight these newly discovered interactors in Fig S1.

Figure S1 was updated according to the reviewer's suggestion. Novel interactions are now presented on a grey square background. The legend for Figure S1A on Page 26, line 5, has been updated to: "(A) Dot-plot visualization of the Commander complex proteins' interactors detected by AP-MS. Each node color corresponds to the abundance of the average spectral count for each prey, and the node size indicates the relative abundance of the prey. BFDR values are denoted by circles around the nodes, and novel interactions are highlighted with a grey background."

Line 567. Give protein concentration in the BS3 cross-linking reaction.

As the cross-linking was conducted during ultrafiltration, the protein concentration increased during the crosslinking process. Therefore, we cannot specify the exact protein concentration. The 2 mM BS3 concentration we used was recommended in a protocol provided by the manufacturer.

FigS2. Add title "cross-linked" above panels D-G. Show FSC curves for the final maps.

The requested titles were added and FSC curves plotted for the final reconstructions as requested.

Line 596. Should "tandem" be replaced with "in parallel"? Tandem implies sequential order.

The text was modified according to reviewer's suggestion.

Fig3J. Indicate the NN-CH domain in the figure. Should CCDC93 NTD be changed to CCDC93 NN-CH?

We have changed the labeling to improve the distinction between NTDs of COMMD proteins and the NN-CH domains of CCDC proteins. These changes have also been implemented in the text by replacing CCDC NTDs with NN-CH.

FigS4. Fix panel assignment (F panel is skipped). The comparison to Healy et al 2023 AF2 model is interesting and will be clearer using pipes and plunks depiction or "ragdoll model" with complex parts presented as geometrical shapes to highlight dramatic differences in COMMDs ring rotation and CCDCs coils. Aligning the models on Vps35L may be more informative.

We thank the reviewer for pointing out the error in panel assignment and have rectified the issue. We settled on pipes representation for the comparison as it clarified the visualization significantly. While we did align the coils via VPS35L to test the representation, we felt that the current alignment center at DENND10 provides a clearer distinction on the relative differences in overall conformation of Retriever vs. the COMMD ring.

Line 134. DENND10 binds the commander via CCDCs, making the statement in line 134 counterintuitive. Please clarify this point.

We previously wrote: "Furthermore, VPS35L and DENND10 are predominantly associated with the complex, while the association of the CCDCs appears weaker," which is indeed confusing. The clustering to find relative similarities takes into account the relative amount of each prey in each sample, and compares that to the other samples. As CCDCs have interactions that are different from the rest of the Commander complex proteins, their association to the other proteins is weaker despite their physical proximity to the COMMD proteins in the Commander complex. The text on Page 4, line 12 was modified to "Furthermore, VPS35L and DENND10 are predominantly associated with the complex, while the association of the CCDCs appears weaker due to their interactions different from the rest of the complex (Fig. 1C)."

Boesch DJ et al. (2023) Structural Organization of the Retriever-CCC Endosomal Recycling

Complex. *bioRxiv*. doi:10.1101/2023.06.06.543888

Healy MD et al. (2023) Structure of the endosomal Commander complex linked to Ritscher-Schinzel syndrome. *Cell* 186:2219-2237 e2229. doi:10.1016/j.cell.2023.04.003

Reviewer #2:

Remarks to the Author:

Laulumaa et al. here report the use of AP-MS and BioID-MS to make a comprehensive characterization of stable and transient interactions of the human Commander complex. The authors propose a set of biological processes where Commander could have essential roles such as endosomal transport, actin nucleation, immune response, transcription regulation, centriole replication, centrosomal targeting and cilium assembly. In addition, the authors purified and solved the structure of the whole Commander complex by single particle cryo-EM. The structure consist of a 16-protein assembly arranged in two sub-complexes; an heterodecameric ring of the COMMD proteins (COMMD 1-10) connected to the retriever sub-complex (VPS35L, VPS26C and VPS29) by the CCDC22-CCDC93 heterodimer which interacts with DENND10 through a V-shaped coil (R-coil). The atomic model built into the cryo-EM density is supported by several XL-MS crosslinks. Overall, this is a well-executed study that provides the molecular architecture of the full Commander complex and spans its interactome. Unfortunately, the recent publication of the structure of the Commander complex (Healy et al. 2023) diminishes the originality of the present work. Nevertheless, while the structure of the Commander complex from Healy et al. was determined by combining data form cryo-EM, X-ray crystallography and AF modelling of distinct sub-structures, the present work by Laulumaa et al. provides a complete picture of the full sixteen subunits. Sadly, the authors have not done a clear comparative analysis between both structures putting an emphasis on aspects that were not presented/discussed by Healy et al. On the other hand, the interactome analysis reads as a dry catalog of interactions with assigned GO terms and biological processes. The manuscript lacks a logical flow where functional/biological mechanisms are addressed throughout the structure. Similarly, the lack of functional assays 'in cellulo' diminishes general interest. For example, assessing the integrity of the Commander complex throughout site directed mutagenesis and evaluating the significance on novel roles such as cilia modulation could add novelty and interest to a wider audience. In summary, the lack of novelty in the structure and the absence of functional implications connected to Commander assembly have dumped the excitement for this manuscript.

Regarding the reviewer's comment "*Sadly, the authors have not done a clear comparative analysis between both structures putting an emphasis on aspects that were not presented/discussed by Healy et al.*", we would like to justify the lack of explicit comparisons to the Healy et al. structure. Their model's quaternary structure is based on predictions by alphafold2, while ours is based on experimental evidence combined with predictions. For the substructures that both models have experimental evidence on, our models agree. Therefore, we feel that extensive comparisons between the models is not relevant in the context of the present manuscript.

Other more specific issues include:

- This reviewer has not been able to see the PDB validation report to assess the PDB model.

We apologize and acknowledge that the reports were only provided during the review process for the editor and reviewers.

- Supplementary Table (REAGENT or RESOURCE) lacks several EMD and PDB codes. They are denoted as 'EMD-XXX' or 'PDB: XXXX'. Same in 'Data availability' (lines 710-718)

We have rectified this error in the manuscript on Page 17, lines 35-40, and thank the reviewer for pointing out the missing codes.

- There is a very early fall of in the FSC(unmasked). Could it be because of the very large box size used in relation to the size of the complex?. Was it intended to capture flexible regions?.

We acknowledge this feature in the FSC curves and as the Reviewer suspects it is due to the relatively large box size used here (required for not cropping out flexible parts of the complex). To demonstrate the effect of box size on FSC(unmasked), we have prepared cropped half-maps and recalculated the unmasked FSC, presented below (Figures 1 and 2). Please note that the "corrected" curves give highly similar results as expected and these values are the reported ones as they take this issue with box size into account.

Figure 1. Gold standard Fourier shell correlation plot for the consensus map of Commander complex at box size of 400 px.

Figure 2. Gold standard Fourier shell correlation plot for the consensus map of Commander complex at box size of 560 px (original box size).

- In section 4.2 of the validation report, there is a big difference in the reported resolution and the unmasked-calculated resolution. Could this be related to the large box size used?

Yes. See above.

- It might be something odd from the validation report but the mask looks larger in 2.6.1 than the projection in 2.1 (although the map in 2.5.1 looks large like the mask, it looks different to the 2.5.2).

We have carefully checked the map and mask visualizations in the validation report, section 6 (map visualization). We note that the maps are shown at different scale in 6.5.1 (primary map) and 6.5.2 (raw map). This is a feature of the PDB validation server and beyond our control. The mask visualized in 6.6.1 is visualized in the same scale as the primary map. We confirm that these are how maps are displayed in the validation reports and do not affect our results or the maps we have deposited.

The following additional modifications were made to the manuscript:

1)

We added additional experiments to the manuscript to further strengthen the interactome data. We generated HEK cell lines with pathogenic point mutations associated with Ritscher-Schinzel syndrome in VPS35L and CCDC22, and conducted BioID-MS for those samples. The following text was added to the manuscript on Page 10, line 3: "Ritscher-Schinzel syndrome point mutations alter PPIs of CCDC22 and VPS35L.

The Commander complex has been associated with Ritscher-Schinzel syndrome (RSS) or X-linked intellectual disability (XLID) via point mutations in VPS35L and CCDC22 [PMID:31712251; PMID:21826058; PMID:23563313; PMID: 24916641]. Disease variants CCDC22(T17A), CCDC22(Y557C), and VPS35L(A830T) (Fig. 6A) are listed as "pathogenic" for RSS in GnomAD database [PMID:32461654]. Cell lines expressing these disease variants were generated for BioID-MS analysis to investigate their effect at the PPI level (Data S1).

VPS35L(A830T) was shown to abolish its interaction with VPS29 [PMID:31712251]. Our data shows that the A830T mutation does not inhibit interaction with VPS26C, but separates VPS35L from the rest of the Commander complex and disrupts its interaction with the WASH complex (Fig. 6A-B). The RSS disease variants of CCDC22 have weaker interactions to COMMD proteins compared to the wild-type, but interact more with the WASH complex (Fig. S6B). The Reactome pathway analysis of disease mutant specific PPIs shows the strongest enrichment of R-HSA-5617833.4~Cilium assembly pathway for CCDC22(T17A), and R-HSA-6811440.2~Retrograde transport at the Trans-Golgi-Network for both variants (Fig. S6E)."

2)

In the discussion section on Page 11 line 7, we added "Interestingly, our finding that besides disrupting the Commander complex assembly, the RSS variant VPS35L(A830T) also loses its affinity to the WASH complex. This is consistent with the presented prediction for potential WASH binding interface, where VPS29 is located at the centre."

3)

The following paragraph was added to the discussion on Page 12, line 9: "We used point mutants CCDC22(T17A), CCDC22(Y557C), and VPS35L(A830T) to investigate the disease mechanism of RSS at the PPI level. The VPS35L(A830T) mutation disrupted the Commander complex assembly, and completely blocked interaction to WASH complex, whereas RSS mutations shifted the HCIs of CCDC22 from the Commander complex towards retrograde transport and Golgi trafficking. This implies a complex molecular etiology of RSS."

4)

The following paragraph was added to Materials and methods on Page 12, line 31: "Generation of RSS disease variants. RSS disease point mutations were introduced to CCDC22 and VPS35L genes by site-directed mutagenesis using Q5® High-Fidelity DNA Polymerase (NEB #M0491) and the following primers: 5'-GGCGCGGCAGTTCCT and 3'-AACTGCCGCGCCGGC (CCDC22(T17A)), 5'-AAGGCCTGTAAGTATCTAGCTGC and 3'-GATACTTACAGGCCTTCCGAACA (CCDC22(Y557C)), and 5'-TCCACCATGAGCCAGGAG and 3'-CTCATGGTGGAGAGGAGATGC (VPS35L_A830). The disease variants and wild-type genes were cloned into C-terminal ultraID containing MAC3-tagged vectors [PMID:35384245]."

5)

Following update was made to Affinity purification protocol on Page 13, line 9: "An additional 50 µM of biotin was added for proximity labelling (BioID) for 24 h (MAC-tagged constructs) or 5 hours (MAC3-tagged constructs)."

6)

Figure S6 legend on Page 28, line 14 was updated to: "Fig. S6. Molecular interactors, context, and cellular pathways connected with individual Commander complex components, related to Figure 5. (A) Dot-plot visualization (BFDR \leq 0.05) of interactors of the Commander complex detected by the BioID-MS. Node color corresponds to the abundance of the average spectral count for each prey, and node radius to its relative abundance. (B) Dot-plot visualization of RSS syndrome related point mutants of CCDC22 analyzed by BioID-MS. All PPIs passing HCI criteria to any of the CCDC22 variants are plotted, with HCIs are indicated with black outline and non-HCIs with light blue. Node color corresponds to the bait normalized abundance of the average spectral count for each prey, and node radius to its relative abundance across all baits determined by ProHits-Viz. (C) Reactome pathways enriched for the Commander complex proteins. (D) Molecular level localization of the Commander complex proteins obtained by MS-microscopy. (E) Reactome pathways enriched ($p < 0.005$, values marked in bars) for the RSS disease variant HCIs distinct from the wild-type CCDC22."

7)

Figure 6 legend on Page 26, line 35 was updated to: "Fig. 6. RSS and XLID related mutations and putative interaction interfaces of the Commander complex. (A) Three mutations associated with RSS or XLID are highlighted within the context of the Commander complex structure. (B) Effect of A830T mutation on VPS35L in BioID-MS. All PPIs passing HCI criteria to either wild-type VPS35L or VPS35L(A830T) are plotted for both constructs, with HCIs indicated using black outline and non-HCIs with light blue outline. Node color corresponds to the bait normalized abundance of the average spectral count for each prey, and node radius to its relative abundance across all baits determined by ProHits-Viz (C) Composite model of the Commander complex,

indicating putative interaction interfaces with tubulin polyglutamylase complex (TPGC). (D) Rotated view of the model in (C), with putative interaction interface of the WASH complex indicated."

8)

Sentence on Page 11, line 44 "We did not detect any Rab proteins as PPIs of any of the Commander complex proteins, supported by the decoupling of Rab7 function from Retriever" was changed to "We did not detect any Rab proteins as HCIs of the Commander complex proteins except for Rab9a in the RSS variants of CCDC22. This observation aligns with the known decoupling of Rab7 function from the Retriever."

Decision Letter, first revision:**Message:** 28th Sep 2023

Dear Professor Varjosalo,

Thank you for submitting your revised manuscript, "Structure and Interactions of the Endogenous Human Commander Complex". After careful consideration and discussion with my colleagues, I am sorry to have to tell you that we do not feel that the referees' comments have been sufficiently addressed to justify sending this revision back for peer review.

This unusual course of action is taken occasionally to avoid unproductive rounds of review that result in reviewer fatigue and damage the chances of the manuscript obtaining a fair and objective evaluation. Such situations are not in an author's best interest so we try to avoid them when it seems prudent to do so.

In order to consider this manuscript further we would request that you please do your best to fully address all of the comments of the reviewers, as well as our editorial guidance. In particular, please do make an effort to visually compare the models (experimental or predicted) of the complex, as well as add further discussion to the text. Both reviewers pointed out this to be an issue with the previous version of the manuscript, and editorially, we agree with their comments. This will be important for the readers to fully understand how this study compares with the literature, and will increase impact and accessibility of your work.

We would kindly ask that in the revised manuscript, you clearly state the differences in the native complex obtained in this study, and how it stands compared to published literature. Please revise both the manuscript and the point-by-point response to address these points.

Should you be able to adequately respond to these and the reviewers' other concerns, we would be happy to look at a revised manuscript again.

We shall hope to receive your revised version as soon as possible. If you anticipate a delay of more than four weeks, however, please let us know. We will be happy to consider your revision so long as nothing similar has been accepted for publication at Nature Structural & Molecular Biology or published elsewhere. Should your manuscript be substantially delayed without notifying us in advance and your article is eventually published, the received date may be that of the revised, not the original, version.

Nature Structural & Molecular Biology is committed to improving transparency in authorship. As part of our efforts in this direction, we are now requesting that all authors identified as 'corresponding author' on published papers create and link their Open Researcher and Contributor Identifier (ORCID) with their account on the Manuscript Tracking System (MTS), prior to acceptance. This applies to primary research papers only. ORCID helps the scientific community achieve unambiguous attribution of all scholarly

contributions. You can create and link your ORCID from the home page of the MTS by clicking on 'Modify my Springer Nature account'. For more information please visit www.springernature.com/orcid.

If you are not interested in submitting a suitably revised manuscript in the future please let me know immediately so we can close your file. If you have any questions, please contact me.

Please use the link below to submit a suitably revised manuscript and updated response to referees when they are ready.

[Redacted]

Sincerely,

Katarzyna Ciazynska
(she/her)
Associate Editor
Nature Structural & Molecular Biology
<https://orcid.org/0000-0002-9899-2428>

Author Rebuttal, first revision:

We thank the reviewers for detailed and positive comments on our manuscript "Structure and Interactions of the Endogenous Human Commander Complex". See below our point-by-point response to the comments (in gray italics). We have expanded our comparison to the existing model of Healy et al. by including two new main figures and adding new material to the supplementary figures. We have also included two pages worth of text on the comparisons, to highlight where the two models differ. Furthermore, we have included additional experiments on the disease mutations, not requested by the reviewers. We feel that these experiments complement well the other comparison we have added on the differences between these two models. These and other additional modifications are reported at the end of this rebuttal.

Reviewer 1:

Remarks to the Author:

Recently discovered Commander complex exerts its function in multiple processes; however, structural information is lacking to deconvolute the underlying mechanisms. The proposed manuscript elegantly combines mass-spectrometric analysis with cryo-electron microscopy to visualize the whole commander complex and systematically analyze its interactome. This approach is granted for a 16-subunit commander complex and, in the broader context, sets up a promising way to parallelize molecular and structural discoveries of challenging biological assemblies.

The authors first created cell lines with one of the commander subunits with MAC tag (BirA* biotinylation enzyme plus affinity purification peptides). They systematically explored subunits interactomes, cross-referencing Bio-ID and affinity purification mass spectroscopy (AP-MS) to get an insight into the stability and composition of commander assembled from endogenously expressed properties (except the bait under control of inducible promotor). The verified reported and discovered a plethora of new binders. The binders list highlights multiple roles of the commander. It includes proteins working in the Cilium Assembly pathway, membrane trafficking (exocyst, PI3P phosphatase), inflammation, and cytoskeleton organization, to name a few functional clusters. Finally, the MS analysis pinpointed posttranslational modification sites.

Following the MS, the cell line with COMMD9 bait was used to isolate the natively assembled commander for cryo-EM structure determination. SPA analysis converged on the map resolving primarily the COMMD ring. Cross-linking was used to restrict the conformational heterogeneity of the complex and succeeded in resolving the whole commander complex. As expected, the COMMD ring map still led in resolution and resolved side chains; nevertheless, the CCDCs and Retriever modules of the commander could now be resolved to the secondary structure using focused alignment. Commander architecture from cryo-EM map drastically differs from the predicted models (Healy et al. 2023; Boesch et al. 2023), featuring ~90 degrees of the COMMD ring rotation and other major differences. This finding is also valuable as a benchmark for predicting complex tertiary structures. Furthermore, the structural studies were complemented by molecular dynamic simulations pinpointing conformation variability in COMMD that may bridge retriever and DENND10. Finally, the manuscript looks closely at the loosely associated interactors discovered by proximity biotinylation (BioID-MS). BioID-MS proteome is dominated by inter-complex binding reiterating on commander multiple roles. Distance dependence of biotinylation by

BirA allowed the identification of two interaction clusters in the commander surface.*

Overall the manuscript brings important novel insights and is written compellingly with sufficient experiential evidence. Authors deserve special credit for clear visual language and concise text.

Here I propose text and figure changes to improve manuscript clarity further.

Figure 1

Fig1A is a good entry illustration but can be made more explicit. I suggest adding a legend for the elements of MAC tag and affinity resins that allow understanding of which tag element is used, similar to the figures in your Liu et al 2018 paper. The arrows from AP-MS and XL-MS branches point to a mass spectrometer (?), but the BioID-MS - to the PPI analysis icon. I suggest leaving only PPI analysis in and converging on it all three paths. Outline conditions for harsh and soft lysis conditions in the figure, i.e. "harsh, 0.1% SDS", "soft, 0.5% IGEPAL".

Thank you for your feedback on Figure 1.

In response to your suggestions, we have revised Fig 1A to enhance its clarity and understandability. We added labels for both the SH-tag and the biotin ligase BirA, aiming to provide clearer visualization of the involved elements.

We would like to clarify the representation of the Mass spectrometer and PPI analysis in Fig 1A. They are intended to represent the same analytical endpoint. To address the inconsistency you pointed out, we've adjusted the figure such that the arrow from BioID-MS now correctly points to the combined MS-analysis and PPI analysis icon. This should now provide a consistent representation of the analytical endpoints for the different techniques.

Regarding the 'lysis conditions', we appreciate your suggestion on detailing the lysis conditions directly in the figure. However, the distinction between soft and harsh lysis is nuanced, involving more than just the choice of detergent. Specifically, the harsh lysis procedure also incorporates sonication in the presence of benzonase. We believe that these intricacies are best detailed in the experimental section, where they can be elaborated upon more extensively. Hence, we've opted to maintain a streamlined

representation in Fig 1A, without the specific detergent details, to ensure its simplicity and accessibility.

Replace the SEC panel in panel A with panel E. Use text instead of pictograms of molecules for SEC peaks designation.

To clarify, the SEC depiction in panel A is intended as a schematic representation, designed to provide an overview of the workflow employed in our study. It is not a direct representation of experimental results.

On the right from the Cryo-EM block, you could name and show all four maps from grey circles in FigS2.

Thank you for your feedback regarding the representation of the Cryo-EM block in our figure.

We understand the value in showcasing detailed results, as you've suggested with the four maps from grey circles in FigS2. However, the primary intent behind Figure 1 panel A was to provide a schematic overview of the entire workflow. Introducing specific results into this schematic might divert from its original purpose of giving a broad-strokes view of the methodology.

Also, please include Coomassie stained PAGE of native and cross-linked Commander used for cryo-EM.

Thank you for your suggestion to include a Coomassie stained PAGE of the native and crosslinked Commander used for cryo-EM.

While in general we appreciate the value of such data, here we've decided against this inclusion for the following critical reasons:

1. The procedure for purifying the crosslinked Commander complex results in a very limited yield (~25 μ l at ~0.1 mg/ml). We, therefore, prioritized cryo-EM grid preparation over PAGE analysis. We were confident in this approach, as we had extensive AP-MS evidence of the constituents of the complex before crosslinking and identified all components of the complex in the MS data we collected.

2. Furthermore, given the mild nature of our crosslinking procedure, we anticipate a set of heterogeneously crosslinked complex isoforms in a PAGE analysis. Such an analysis would primarily indicate that crosslinking has occurred, without providing intricate details. We can obtain more precise information about the crosslinking pattern from the MS data we collected from the crosslinked, gel-filtered complex.

Fig 1D. Define table axes.

Table axes (bait / prey) have now been added to Figure 1 panel D.

If feasible, introduce a better graphical explanation of the colour/circle size code.

Thank you for the feedback on our graphical representation.

To enhance clarity, we've renamed the colour/circle size codes to "Bait normalized AvgSpec" and "Relative AvgSpec." We've expanded on this in the figure legend with the following explanation on Page 26, line 9: "The color of each circle represents the abundance of each prey normalized to the mean abundance of the bait protein, and the circle radius indicates the relative abundance across all samples calculated by ProHits-Viz."

We hope this adjustment provides a clearer understanding of the graphical elements used.

Panel A can be used to reference specific branches of the MS. Consider moving panel B to supplementary if more space is needed.

We have incorporated a reference to panel A in the text.

AvgSpec/Relative AvgSpec abundance needs a better explanation for readers outside the MS field. Consider extending the Quantification and statistical analysis chapter and explaining the interpretation of two marginal data points (i.e. for good and bad binders).

The average spectral count corresponds to the abundance of each prey protein in the sample and is represented by the node color. To evaluate the relative weight of each interaction among the presented samples, we calculated the relative abundance using the ProHits-Viz tool.

We have updated the Figure 1 legend on Page 26, line 9 to read: "The color of each circle represents the abundance of each prey, normalized to the mean abundance of the bait protein, and the circle radius indicates the relative abundance across all samples calculated by ProHits-Viz."

Line 142. Comment why COMMD9 was chosen as bait.

As there was initially very little information about the Commander complex assembly, we selected a few bait proteins for purification optimization. Out of the tested bait proteins, COMMD9 gave the highest yield, and was thus chosen for further optimization.

Line 480. Cloning of Commander complex components. Describe cell genotype in plain language. If I understand correctly, cell lines contained an additional copy (over the WT genome) of genome-integrated MAC-tagged tagged bait under a tetracycline-inducible promoter. Upon induction, the binders will preferentially bind to the tagged due to its higher titer than WT copies. What is the estimated bait expression level compared to the endogenously expressed protein?

The cell line used employs the Flp-In™ system, which allows the insertion of a single copy of the gene of interest at the FRT site in HEK cells. Expression of the protein of interest is induced using tetracycline, resulting in an expression level parallel to and on the same scale as the endogenous protein [PMID:19156129; PMID:23455922; PMID:28330616]. In BioID-MS experiments, proximity labeling biotinylation is induced using biotin. Intact protein complexes (AP-MS) or biotinylated proteins (after disruption of native protein complexes in BioID-MS) are extracted from the cell lysate using Strep-tactin resin, which binds both biotin and the SH-tag. The method is described in more detail in the articles by Liu et al [PMID:29568061], which we reference in the text on Page 3, line 5, and on Page 13, line 27.

Line 91. Highlight these newly discovered interactors in Fig S1.

Figure S1 was updated according to the reviewer's suggestion. Novel interactions are now presented on a grey square background. The legend for Figure S1A on Page 28, line 20, has been updated to: "(A) Dot-plot visualization of the Commander complex proteins' interactors detected by AP-MS. Each node color corresponds to the abundance of the average spectral count for each prey, and the node size indicates the relative abundance of the prey. BFDR values are denoted by circles around the nodes, and novel interactions are highlighted with a grey background."

Line 567. Give protein concentration in the BS3 cross-linking reaction.

As the crosslinking was conducted during ultrafiltration, the protein concentration increased during the crosslinking process. Therefore, we cannot specify the exact protein

concentration. The 2 mM BS3 concentration we used was recommended in a protocol provided by the manufacturer.

FigS2. Add title "cross-linked" above panels D-G. Show FSC curves for the final maps.

The requested titles were added and FSC curves plotted for the final reconstructions as requested.

Line 596. Should "tandem" be replaced with "in parallel"? Tandem implies sequential order.

The text was modified according to reviewer's suggestion.

Fig3J. Indicate the NN-CH domain in the figure. Should CCDC93 NTD be changed to CCDC93 NN-CH?

We have changed the labeling to improve the distinction between NTDs of COMMD proteins and the NN-CH domains of CCDC proteins. These changes have also been implemented in the text by replacing CCDC NTDs with NN-CH.

FigS4. Fix panel assignment (F panel is skipped). The comparison to Healy et al 2023 AF2 model is interesting and will be clearer using pipes and plunks depiction or "ragdoll model" with complex parts presented as geometrical shapes to highlight dramatic differences in COMMDs ring rotation and CCDCs coils. Aligning the models on Vps35L may be more informative.

We thank the reviewer for pointing out the error in panel assignment and have rectified the issue. A "ragdoll model" using geometric shapes as suggested has now been included

as new Figure 5. We have carried out the alignment of Healy et al. model on our model in two different ways to highlight the differences from multiple angles in Figure 5:

"A simplified "ragdoll" representation of major components of the Commander complex **(A)** from this study and **(B-D)** the overall structural model from Healy et al. highlight the major differences between these models with alignment centers of the models located at the **(B)** COMMD-ring, **(C)** V-coil, and **(D)** DENND10. The COMMD-ring is represented by a disc aligned to the COMMD domains, the coiled-coil domains are represented by cylinders (I and R-coils), or a trapezoidal prism (V-coil + CCDC22 NN-CH), DENND10 as a cylinder and Retriever subcomplex as spheres (VPS29, VPS26C + N-terminal half of VPS35L α -solenoid) or a cylinder (VPS35L C-terminal half of VPS35L α -solenoid). Component relative rotation angles are calculated based on the underlying atomic coordinates of backbone C α atoms."

Line 134. DENND10 binds the commander via CCDCs, making the statement in line 134 counterintuitive. Please clarify this point.

We previously wrote: “Furthermore, VPS35L and DENND10 are predominantly associated with the complex, while the association of the CCDCs appears weaker,” which is indeed confusing. The clustering to find relative similarities takes into account the relative amount of each prey in each sample, and compares that to the other samples. As CCDCs have interactions that are different from the rest of the Commander complex proteins, their association to the other proteins is weaker despite their physical proximity to the COMMD proteins in the Commander complex. The text on Page 4, line 11 was modified to “Furthermore, VPS35L and DENND10 are predominantly associated with the complex, while the association of the CCDCs appears weaker due to their interactions different from the rest of the complex (**Fig. 1C**).”

Boesch DJ et al. (2023) Structural Organization of the Retriever-CCC Endosomal Recycling Complex. bioRxiv. doi:10.1101/2023.06.06.543888

Healy MD et al. (2023) Structure of the endosomal Commander complex linked to Ritscher-Schinzel syndrome. Cell 186:2219-2237 e2229. doi:10.1016/j.cell.2023.04.003

Reviewer #2:

Remarks to the Author:

Laulumaa et al. here report the use of AP-MS and BioID-MS to make a comprehensive characterization of stable and transient interactions of the human Commander complex. The authors propose a set of biological processes where Commander could have essential roles such as endosomal transport, actin nucleation, immune response, transcription regulation, centriole replication, centrosomal targeting and cilium assembly. In addition, the authors purified and solved the structure of the whole Commander complex by single particle cryo-EM. The structure consist of a 16-protein assembly arranged in two sub-complexes; an heterodecameric ring of the COMMD proteins (COMMD 1-10) connected to the retriever sub-complex (VPS35L, VPS26C and VPS29) by the CCDC22-CCDC93 heterodimer which interacts with DENND10 through a V-shaped coil (R-coil). The atomic model built into the cryo-EM density is supported by several XL-MS crosslinks. Overall, this is a well-executed study that provides the molecular architecture of the full Commander complex and spans its interactome. Unfortunately, the recent publication of the structure of the Commander complex (Healy et al. 2023) diminishes the originality of the present work. Nevertheless, while the structure of the Commander complex from Healy et al. was determined by combining data form cryo-EM, X-ray crystallography and AF modelling of distinct sub-structures, the present work by Laulumaa et al. provides a complete picture of the full sixteen subunits. Sadly, the authors have not done a clear comparative analysis between both structures putting an emphasis on aspects that were not presented/discussed by Healy et al. On the other hand, the interactome analysis reads as a dry catalog of interactions with assigned GO terms and biological processes. The manuscript lacks a logical flow where functional/biological mechanisms are addressed throughout the structure. Similarly, the lack of functional assays 'in cellulo' diminishes general interest. For example, assessing the integrity of the Commander complex throughout site directed mutagenesis and evaluating the significance on novel roles such as cilia modulation could add novelty and interest to a wider audience. In summary, the lack of novelty in the structure and the absence of functional implications connected to Commander assembly have dumped the excitement for this manuscript.

Regarding the reviewer's comment "*Sadly, the authors have not done a clear comparative analysis between both structures putting an emphasis on aspects that were not*

presented/discussed by Healy et al.", we have now expanded our comparative analysis by including two new main figures detailing the differences at the major structural region level (**Fig. 5**) as well as within these structural regions (**Fig. 6**).

The following text was added to the results starting from Page 8, line 13:

"Recently, Healy et al. published an integrated structural model for the Commander complex [PMID:37172566]. As the model differs significantly in overall arrangement from the structure presented in this study (**Fig. S4G**), we compared the structures first by superposing the Healy et al. model using different alignment centers to our model (**Fig. 5**). The CCDC scaffolding is similar in both complexes, and the COMMD-ring, DENND10, and the Retriever subcomplex are located in similar positions along it (**Fig. S4G**). Three major differences between the models were found (**Fig 5**): (i) the overall structure is more compactly packed in our model than in the Healy et al. model. (ii) The orientation of the COMMD-ring relative to the CCDC scaffolding is different so that in Healy et al. model the COMMD-ring lacks contact to I-coil which is evident both in our cryo-EM and XL-MS data (**Fig. 2A-B, Fig. S4D**). Notably, the Healy et al. model is incompatible with the crosslink between CCDC93 and COMMD7 detected in this study. Furthermore, the relative orientation between COMMD-ring and DENND10 or V-coil differ by 76° and 117°, respectively. (iii) The twistedness of the scaffolding is dissimilar such that the relative orientation of DENND10 and Retriever differs by 65°.

Healy et al. compiled the overall model of the Commander complex using AF2 combined with X-ray crystallography and cryo-EM data from certain regions of the complex. When compared to our structure, models of the COMMD-ring align well as both studies base the structural models on high-resolution cryo-EM data (**Fig. 6B**). CCDC22 helices $\alpha 15$ and $\alpha 16$, and the HLH-motif of CCDC93 are placed differently in the two models. They are absent in the cryo-EM structure by Healy et al. (EMD-28827, PDB ID 8F2U) whereas their placement in our model is supported by cryo-EM density (**Fig. S4H**). The conformation of DENND10, I-coil, and R-coil is based heavily on AF2 prediction in both models, as our cryo-EM reconstruction has limited resolution in this region (**Fig. S5D**), and the Healy et al. model is entirely based on AF2 prediction for this part (**Fig. 6C**). The overall folds are similar, except that an interaction between the N-lobe of DENND10 and I-coil presented in Healy et al. model (indicated with an asterisk, **Fig. 6C**) is not featured in our model. This may be explained by conformational heterogeneity, as evidenced by our 3DVA analysis of this region (**Fig. 4F**).

In our model, the Retriever subcomplex extends out of the main body of the complex (**Fig. 2E-F**). The tip extension of VPS35L binds VPS26C, while in the Healy et al. model it forms an interaction surface with the CCDC22 part of V-coil (**Fig. 6D**). This interaction seen in the Healy et al. model was predicted by AF2 and may reflect conformational heterogeneity in this region. Finally, Healy et al. solved the crystal structure of VPS29 with VPS35L (24-38) peptide (PDB ID 8ESE), which is consistent with our cryo-EM structure (**Fig. 6E**)."

Other more specific issues include:

- This reviewer has not been able to see the PDB validation report to assess the PDB model.

We apologize and acknowledge that the reports were only provided during the review process for the editor and reviewers.

- Supplementary Table (REAGENT or RESOURCE) lacks several EMDB and PDB codes. They are denoted as 'EMD-XXX' or 'PDB: XXXX'. Same in 'Data availability' (lines 710-718)

We have rectified this error in the manuscript on Page 19, lines 7-12, and thank the reviewer for pointing out the missing codes.

- There is a very early fall of in the FCS(unmasked). Could it be because of the very large box size used in relation to the size of the complex?. Was it intended to capture flexible regions?.

We acknowledge this feature in the FSC curves and as the Reviewer suspects it is due to the relatively large box size used here (required for not cropping out flexible parts of the complex). To demonstrate the effect of box size on FSC(unmasked), we have prepared cropped half-maps and recalculated the unmasked FSC, presented below (Response Letter Figures 1 and 2). Please note that the "corrected" curves give highly similar results as expected and these values are the reported ones as they take this issue with box size into account.

Response Letter Figure 1. Gold standard Fourier shell correlation plot for the consensus map of Commander complex at box size of 400 px.

Response Letter Figure 2. Gold standard Fourier shell correlation plot for the consensus map of Commander complex at box size of 560 px (original box size).

- In section 4.2 of the validation report, there is a big difference in the reported resolution and the unmasked-calculated resolution. Could this be related to the large box size used?

Yes. See above.

- It might be something odd from the validation report but the mask looks larger in 2.6.1 than the projection in 2.1 (although the map in 2.5.1 looks large like the mask, it looks different to the 2.5.2).

We have carefully checked the map and mask visualizations in the validation report, section 6 (map visualization). We note that the maps are shown at different scale in 6.5.1 (primary map) and 6.5.2 (raw map). This is a feature of the PDB validation server and beyond our control. The mask visualized in 6.6.1 is visualized in the same scale as the

primary map. We confirm that these are how maps are displayed in the validation reports and do not affect our results or the maps we have deposited.

The following additional modifications were made to the manuscript:

1)

We added additional experiments to the manuscript to further strengthen the interactome data. We generated HEK cell lines with pathogenic point mutations associated with Ritscher-Schinzel syndrome in VPS35L and CCDC22, and conducted BioID-MS for those samples. The following text was added to the manuscript on Page 10, line 41: “The Commander complex has been associated with Ritscher-Schinzel syndrome (RSS) or X-linked intellectual disability (XLID) via point mutations in VPS35L and CCDC22^{13,31,70,71}. Disease variants CCDC22(T17A), CCDC22(Y557C), and VPS35L(A830T) (**Fig. 2D**, **Fig. 8A**) are listed as “pathogenic” for RSS in GnomAD database 72. Cell lines expressing these disease variants were generated for BioID-MS analysis to investigate their effect at the PPI level (Data S1).

The VPS35L(A830T) has been suggested to abolish its interaction with VPS29 70. Our data shows that the A830T mutation does not inhibit interaction with VPS26C, but separates VPS35L from the rest of the Commander complex and disrupts its interaction with the WASH complex (**Fig. 8A-B**).

With the CCDC22 RSS disease variants, we discovered less interactions to COMMD proteins (especially to COMMDs 3, 6, and 7), whereas interactions with the WASH complex become pronounced (**Fig. S6B**). Using the Reactome pathway analysis on the CCDC22 disease mutant interactors, we could detect enrichment of ‘Cilium assembly pathway’ for CCDC22(T17A), and ‘Retrograde transport at the Trans-Golgi-Network’ for both variants (**Fig. S6E**). The CCDC22(Y557C) mutation is situated at the tip of the CCDC22 part of V-coil, a region predicted to interact with VPS35L by Healy et al. Surprisingly, our BioID data shows no major changes in the interactome that could be expected if this interaction was significant. On the other hand, the distal location of VPS26C, and thus the tip of VPS35L, from the V-coil is supported by our BioID data where VPS26C and CCDCs are not in close proximity (**Fig. 7B**). However, possible effects on the VPS35L-V-coil interaction caused by this mutation need to be experimentally interrogated.”

2)

The disease mutations were further discussed on Page 12 line 2, “Intriguingly, aside from its effect on Commander complex assembly, the RSS variant VPS35L(A830T) also displays reduced affinity for the WASH complex. The mutation site is located at the

interface between VPS35L and VPS29, and therefore this observation fits well with our proposed WASH binding interface model where VPS29 is located at the center. In contrast, while CCDC22 variants (T17A and Y557C) are also in close proximity to the putative WASH binding site, they have the opposite effect on WASH complex interactions, implying a complex molecular etiology of RSS.”

- 3)

The following paragraph was added to Materials and methods on Page 13, line 30: "RSS disease point mutations were introduced to CCDC22 and VPS35L genes by site-directed mutagenesis using Q5® High-Fidelity DNA Polymerase (NEB #M0491) and the following primers: 5'-GGCGCGGCAGTTCCT and 3'-AACTGCCGCGCCGGC (CCDC22(T17A)), 5'-AAGGCCTGTAAGTATCTAGCTGC and 3'-GATACTTACAGGCCTTCCGAACA (CCDC22(Y557C)), and 5'-TCCACCATGAGCCAGGAG and 3'-CTCATGGTGGAGAGGAGATGC (VPS35L_A830). The disease variants and wild-type genes were cloned into C-terminal ultraID containing MAC3-tag vector [PMID:35384245]."
- 4)

Following update was made to Affinity purification protocol on Page 14, line 3: "An additional 50 µM of biotin was added for proximity labelling (BioID) for 24 h (MAC-tagged constructs) or 5 hours (MAC3-tagged constructs)."
- 5)

The following paragraph was updated on Page 12, line 24: "The top half of the Commander complex contains flexible components (NTDs of COMMDs 1, 7, 9, and 10; HLH-motif of CCDC93), and the bottom half exhibits compositional and conformational heterogeneity, particularly within the Retriever subcomplex. This may reflect physiological assembly and/or the function of the complex. Indeed, COMMD proteins and the CCDCs have been proposed to form a complex without Retriever, termed the CCC-complex 75. Interestingly, assuming a similar head-to-head dimerization mode as Retromer, the structure of Commander permits binding of the CCC-complex on a Retriever dimer without obvious steric clashes (**Fig. S5F-H**). Such analysis relies on having access to complete native structure featuring external surfaces and exposed domains facilitating biological functions in the cellular context. This highlights the need for experimental data in addition to in silico predictions for quaternary structure analysis of large macromolecular complexes (**Fig. 5, Fig. 6**)."
- 6)

Sentence on Page 13, line 2 "We did not detect any Rab proteins as PPIs of any of the Commander complex proteins, supported by the decoupling of Rab7 function from Retriever" was changed to "We did not detect any Rab proteins as HCIs of the

Commander complex proteins except for Rab9a in the RSS variants of CCDC22. This observation aligns with the known decoupling of Rab7 function from the Retriever.”

7)

Figure S6 legend on Page 29, line 30 was updated to: “**Fig. S6. Molecular interactors, context, and cellular pathways connected with individual Commander complex components, related to Figure 5.** (A) Dot-plot visualization (BFDR ≤ 0.05) of interactors of the Commander complex detected by the BioID-MS. Node color corresponds to the abundance of the average spectral count for each prey, and node radius to its relative abundance. (B) Dot-plot visualization of RSS syndrome related point mutants of CCDC22 analyzed by BioID-MS. All PPIs passing HCI criteria to any of the CCDC22 variants are plotted, with HCIs are indicated with black outline and non-HCIs with light blue. Node color corresponds to the bait normalized abundance of the average spectral count for each prey, and node radius to its relative abundance across all baits determined by ProHits-Viz. (C) Reactome pathways enriched for the Commander complex proteins. (D) Molecular level localization of the Commander complex proteins obtained by MS-microscopy. (E) Reactome pathways enriched ($p < 0.005$, values marked in bars) for the RSS disease variant HCIs distinct from the wild-type CCDC22.”

8)

Figure 8 legend on Page 28, line 7 was updated to: “**Fig. 6. RSS and XLID related mutations and putative interaction interfaces of the Commander complex.** (A) Three mutations associated with RSS or XLID are highlighted within the context of the Commander complex structure. (B) Effect of A830T mutation on VPS35L in BioID-MS. All PPIs passing HCI criteria to either wild-type VPS35L or VPS35L(A830T) are plotted for both constructs, with HCIs indicated using black outline and non-HCIs with light blue outline. Node color corresponds to the bait normalized abundance of the average spectral count for each prey, and node radius to its relative abundance across all baits determined by ProHits-Viz (C) Composite model of the Commander complex, indicating putative interaction interfaces with tubulin polyglutamylase complex (TPGC). (D) Rotated view of the model in (C), with putative interaction interface of the WASH complex indicated.”

9)

Figures 5 and 6 from the previous iteration of the manuscript were renumbered to figures 7 and 8. Figure legends for new figures 5 and 6 were added on Page 27, line 21: **Fig. 5. Analysis of overall tertiary fold of the endogenous Commander complex compared to existing literature.** A simplified “ragdoll” representation of major components of the Commander complex (**A**) from this study and (**B-D**) the overall structural model from Healy et al. highlight the major differences between these models with alignment centers of the models located at the (**B**) COMMD-ring, (**C**) V-coil, and (**D**) DENND10. The COMMD-ring is represented by a disc aligned to the COMMD domains, the coiled-coil domains are represented by cylinders (I and R-coils), or a trapezoidal prism (V-coil + CCDC22 NN-CH), DENND10 as a cylinder and Retriever subcomplex as spheres (VPS29, VPS26C + N-terminal half of VPS35L α -solenoid) or a cylinder (VPS35L C-terminal half of VPS35L α -solenoid). Component relative rotation angles are calculated based on the underlying atomic coordinates of backbone C α atoms.

Fig. 6. Comparative analysis of conformational variation in the Commander complex structure compared to existing literature. (**A**) Overview of the Commander complex structure with location of following panels indicated. Comparison of (**B**) the COMMD-ring, (**C**) DENND10, I-coil, and R-coil region, (**D**) Retriever subcomplex from the structure presented in this study and the overall model presented by Healy et al. (**E**) Comparison of VPS29 with VPS35L (13-37) presented in this study (left) and crystal structure of VPS29 with VPS35L (16-38) peptide (right). Major structural differences are highlighted with yellow, and sources of structural data are indicated for each structure. The three disease mutations analyzed in AP-MS and BioID (**Fig. 8**) are indicated in (**D**).

10)

Figure S4H legend on Page 29, line 17 was changed to: " (**H**) Density supporting the placement of CCDC22 α 15 and α 16. Map (EMD-17340) was low-pass filtered to 7 Å using Bsoft."

Decision Letter, second revision:

Message: Our ref: NSMB-A47767B

15th Nov 2023

Dear Dr. Varjosalo,

Thank you for submitting your revised manuscript "Structure and Interactions of the Endogenous Human Commander Complex" (NSMB-A47767B). It has now been seen by the original referees and their comments are below. The reviewers find that the paper has improved in revision, and therefore we'll be happy in principle to publish it in Nature Structural & Molecular Biology, pending minor revisions to satisfy the referees' final requests and to comply with our editorial and formatting guidelines.

In particular, please note that we consulted reviewer #2 again regarding the PDB validation reports. While they did not indicate major issues, they pointed out that there appears to be a noticeable lack of fitting between the COMM domain-containing protein 1 (Chain A; Molecule 1) and the COMM domain-containing protein 7 (Chain G; Molecule 7) with the map. Please ensure to address this discrepancy and discuss it in the manuscript.

We are now performing detailed checks on your paper and will send you a checklist detailing our editorial and formatting requirements in about 2 weeks. Please do not upload the final materials and make any revisions until you receive this additional information from us.

Sincerely,

Katarzyna Ciazynska, PhD
(she/her)
Associate Editor
Nature Structural & Molecular Biology
<https://orcid.org/0000-0002-9899-2428>

Author Rebuttal, second revision:

Response to Reviewer 1:

We thank the reviewer for the positive comments on our manuscript "Structure and Interactions of the Endogenous Human Commander Complex".

Reviewer 1:

My apologies for the delayed reply. I thank the authors for thoroughly addressing my

suggestions. A new Figure 5 elaborating on the global conformation of commander is a welcome addition. I have no further questions to the manuscript.

Final Decision Letter:

Message: 19th Jan 2024

Dear Dr. Varjosalo,

We are now happy to accept your revised paper "Structure and Interactions of the Endogenous Human Commander Complex" for publication as an Article in Nature Structural & Molecular Biology.

Your paper will be published online soon after we receive proof corrections and will appear in print in the next available issue. You can find out your date of online publication by contacting the production team shortly after sending your proof corrections.

Please note that *Nature Structural & Molecular Biology* is a Transformative Journal (TJ). Authors may publish their research with us through the traditional subscription access route or make their paper immediately open access through payment of an article-processing charge (APC). Authors will not be required to make a final decision about access to their article until it has been accepted. Find out more about Transformative Journals

Sincerely,

Katarzyna Ciazynska, PhD
(she/her)
Associate Editor
Nature Structural & Molecular Biology
<https://orcid.org/0000-0002-9899-2428>